# IL-17$^+$ CD8$^+$ T cell suppression by dimethyl fumarate associates with clinical response in multiple sclerosis

Christina Lückel et al.[#]

IL-17-producing CD8$^+$ (Tc17) cells are enriched in active lesions of patients with multiple sclerosis (MS), suggesting a role in the pathogenesis of autoimmunity. Here we show that amelioration of MS by dimethyl fumarate (DMF), a mechanistically elusive drug, associates with suppression of Tc17 cells. DMF treatment results in reduced frequency of Tc17, contrary to Th17 cells, and in a decreased ratio of the regulators *RORC*-to-*TBX21*, along with a shift towards cytotoxic T lymphocyte gene expression signature in CD8$^+$ T cells from MS patients. Mechanistically, DMF potentiates the PI3K-AKT-FOXO1-T-BET pathway, thereby limiting IL-17 and RORγt expression as well as STAT5-signaling in a glutathione-dependent manner. This results in chromatin remodeling at the *Il17* locus. Consequently, T-BET-deficiency in mice or inhibition of PI3K-AKT, STAT5 or reactive oxygen species prevents DMF-mediated Tc17 suppression. Overall, our data disclose a DMF-AKT-T-BET driven immune modulation and suggest putative therapy targets in MS and beyond.

---

[#]A full list of authors and their affiliations appears at the end of the paper.

Multiple sclerosis (MS) is an inflammatory disease of the central nervous system (CNS) that affects ~2.5 million people worldwide causing neurological disability predominantly in young women. The disease is believed to be mediated by self-reactive T cells which initiate and perpetuate inflammation characterized by perivascular immune cell infiltration, demyelination, neuroaxonal damage and inflammatory lesions[1,2]. In support of this assumption the largest group of genes associating with MS susceptibility is involved in antigen presentation to T cells or in T cell pathways[3–5]. Moreover, both CD4+ and CD8+ T cells are present in MS lesions[1,2,6] and T cell infiltration correlates with the activity of demyelinating lesions[7]. Notably, CD8+ T cells are found in higher frequency than CD4+ T cells[8–10] and primarily among CD8+ T cells large clonal expansions have been reported in active demyelinating MS lesions[11]. IL-17-producing CD8+ (Tc17) are enriched in cerebrospinal fluid (CSF) in early MS[12] and Tc17 frequencies in CSF correlate with disability[13]. Furthermore, increased frequencies of Tc17 cells were detected in peripheral blood (PB) of MS patients as compared to healthy controls[14] and Tc17 cells were present in active areas of acute and in chronic MS lesions alongside with IL-17-producing CD4+ (Th17) T cells[15], implicating a contribution of both subpopulations to MS pathogenesis. Interestingly, many of IL-17-producing CD8+ T cells in MS patients bear features of mucosal associated invariant T (MAIT) cells, which are MHC-related protein 1 (MR1)-restricted CD8+ T cells dependent on commensal microbiota[6,16–20]. Functionally, using experimental autoimmune encephalomyelitis (EAE) as a pre-clinical mouse model for MS, we showed that Tc17 cells provided "reverse help" for the encephalogenicity of IL-17-producing CD4+ T (Th17) cells via their hallmark cytokine IL-17A[12], revealing an important Tc17-dependent enhancement of Th17-mediated autoimmunity of the CNS.

Tc17 cells are induced by the cytokines IL-6 and transforming growth factor (TGF)-β, and require the type 17-related transcriptional regulator RORγt[21]. In contrast to "canonical" cytotoxic T lymphocytes (CTLs), Tc17 cells are non-cytotoxic and express diminished levels of the CTL-specific transcription factors T-BET and EOMES, which counter regulate their differentiation[22–24]. In addition to MS, Tc17 cells are also involved in the pathology of psoriasis[21], an autoimmune disease of the skin.

Dimethyl fumarate (DMF) is an efficient immunomodulatory drug, applied in MS and psoriasis; however, to date the mechanism of its beneficial action has remained unclear[25]. It is known that DMF succinates kelch-like ECH-associated protein 1 (KEAP1), leading to the activation of nuclear factor erythroid 2-related factor 2 (NRF2). Although this pathway is believed to protect astrocytes and neurons by inducing an anti-oxidative response[25], data from NRF2-deficient mice suggested NRF2-independent mechanisms in the anti-inflammatory activity of DMF[26]. Indeed, succination of GAPDH by DMF suppressed aerobic glycolysis in myeloid and lymphoid cells, thereby limiting autoimmunity[27,28]. Finally, succination of the reactive oxygen species (ROS) scavenger glutathione (GSH) reduced its antioxidant capacity thus upregulating endogenous ROS in DCs, tumor cells, monocytes and macrophages[29–32]. In line with these data, an increase in ROS was observed in monocytes and T cells from MS patients upon DMF therapy[32,33]. ROS display concentration-dependent effects on T cells ranging from activation at physiological levels to inhibition of function at sustainably upregulated concentrations[34,35].

Considering the central contribution of T cells to CNS pathology and the efficacy of DMF treatment in MS[36], we hypothesized that DMF may target them. We therefore analyzed cytokine production and molecular changes in T cells in response to DMF therapy in patients and in the mouse model EAE. Our analyses identified Tc17 cells as a target cell population of DMF. We define critical pathways including PI3K-AKT-FOXO1-T-BET and STAT5 leading to histone modifications, which control this process in a GSH-dependent manner. Thus, our data suggest a new rational approach for targeting Tc17 cells in MS and other IL-17-mediated disorders.

## Results

**Tc17 suppression accompanies positive response to DMF in MS.** To understand a relation between frequencies of IL-17-producing CD4+ or CD8+ T cells and a response to DMF, we analyzed 72 cryopreserved peripheral blood mononuclear cell (PBMC) samples isolated from peripheral blood (PB) of a cohort of 36 patients with MS before and after initiation of DMF therapy, which did (responders, $n = 18$) or did not fulfill (non-responders $n = 18$) no evidence of disease activity-3 (NEDA-3) criteria after about one year of treatment (Supplementary Fig. 1a, Supplementary Tables 1–3). NEDA-3 is a clinically relevant composite score reflecting therapeutic efficiency[37] defined as: (i) no relapses, (ii) no sustained disability progression measured with the expanded disability status scale (EDSS) and (iii) no new/enlarging T2-weighted lesions in magnetic resonance imaging (MRI). Baseline samples and samples obtained after DMF therapy originated from the same patients.

For the analysis, frozen PBMCs were thawed and stained (Supplementary Fig. 1a, b). The change in the abundance of IL-17-producing CD8+ T (Tc17) cells before versus after therapy initiation distinguished responders versus non-responders (Fig. 1a). Interestingly, the patients who responded to the therapy showed a significantly lower Tc17 frequency after treatment as compared to the therapy initiation, while in non-responders the frequency of Tc17 cells was not significantly changed. This was in contrast to IL-17-producing CD4+ T (Th17) cells which were not differentially abundant before and after the start of the therapy in responders as well as in non-responders (Fig. 1a). Notably, at the therapy start, responders harbored higher frequencies of Tc17 cells as compared to non-responders, whereas Th17 cell abundance was not significantly different in the analyzed patient cohort (Fig. 1b, Supplementary Fig. 1c). These results suggested inhibition of Tc17 cells as a possible mechanism of DMF therapy. Indeed, DMF suppressed IL-17 in human Tc17 cells cultured in vitro, indicating a direct effect (Supplementary Fig. 1d). This inhibition was likely dependent on ROS, as the ROS-scavenger glutathione (GSH) restored IL-17 production as examined by intracellular staining and ELISA (Supplementary Fig. 1d, e). Likewise, DMF inhibited IL-17 in murine Tc17 cells in a ROS-dependent manner, as GSH, its precursor N-acetyl-L-cysteine (NAC) or Trolox, a vitamin E derivative and GSH-independent ROS scavenger, reversed the altered cytokine production (Fig. 1c and Supplementary Fig. 1f), suggesting a similar regulation of IL-17 in Tc17 cells by DMF across mice and humans.

As expected, DMF mediated a profound depletion of GSH (Fig. 1d) as measured by a significantly reduced GSH/GSSG ratio. This resulted in upregulation of endogenous ROS in Tc17 cells, as detected by staining with CM-H2DCFDA (Fig. 1e), which was reversed by the addition of GSH, confirming that its depletion mediates the upregulation of ROS. Notably, at concentrations of 20 μM, which efficiently suppressed IL-17 production in murine Tc17 cells (Fig. 1c), DMF did not cause appreciable cell death, whereas in line with published reports[33,38], higher concentrations did (Supplementary Fig. 1g). Importantly, addition of GSH reversed the cell death (Supplementary Fig. 1g), suggesting a dose-dependent effect of DMF relating to GSH-depletion, which suppressed IL-17 at moderate levels, whereas at higher triggered cell death. Although

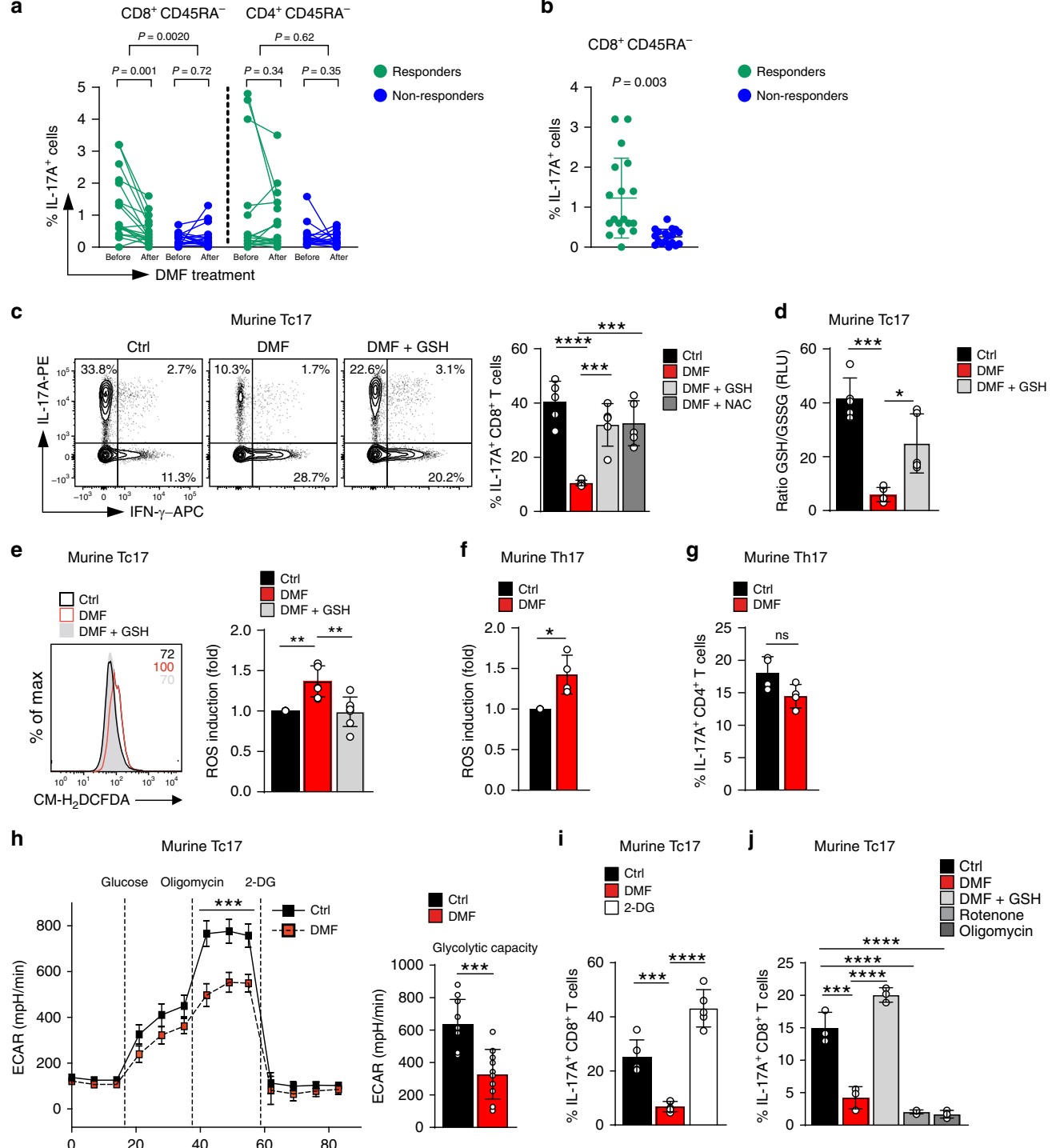

DMF reduced proliferation of Tc17 cells in a GSH-dependent manner, this was not the cause for the reduced IL-17 production, since the frequencies of IL-17-producing cells were reduced in each proliferation cycle (Supplementary Fig. 1h, i).

As IL-23 is important for the pathogenicity of Th17 cells[6], we performed analysis of the DMF effects on Th17 cells cultured under pathogenic conditions including IL-23. Similar to Tc17 cells, DMF treatment led to significant ROS upregulation in pathogenic Th17 cells (Fig. 1f). However, this did not result in a statistically significant reduction of IL-17 (Fig. 1g), suggesting that Tc17 are particularly responsive to DMF-mediated GSH depletion. In line with a recent report showing that DMF inhibits

glycolysis by succination of GAPDH[28], DMF suppressed glycolytic capacity also in Tc17 cells (Fig. 1h). However, IL-17 production by Tc17 cells was rather independent of glycolysis since the inhibitor 2-deoxy-D-glucose (2-DG) upregulated IL-17 while downregulating IFN-γ (Fig. 1i and Supplementary Fig. 1j). In contrast to Th17 cells, which depend on glycolysis[39], IL-17 production by Tc17 cells relied on oxidative phosphorylation (OXPHOS), as the inhibitors rotenone or oligomycin significantly suppressed the IL-17 production (Fig. 1j). Overall these data indicate that DMF preferentially targets Tc17 cells in a ROS- and GSH-dependent manner, probably by a different mechanism as compared to Th17 cells.

**Fig. 1 Suppression of IL-17A production in CD8$^+$ T cells by DMF is ROS-dependent. a** Flow cytometry of IL-17A in CD8$^+$CD45RA$^-$ or CD4$^+$CD45RA$^-$ cells from blood of the same MS patients before and after DMF therapy fulfilling (reponders, $n = 18$), or not (non-responders, $n = 18$) NEDA-3 criteria after treatment (Supplementary Tables 1-3, Supplementary Fig. 1a, b). The observer was blinded to experimental groups. **b** Frequency of CD8$^+$CD45RA$^-$IL-17A$^+$ cells before DMF therapy in responders and non-responders. **c-j** Naive CD62L$^+$CD44$^-$CD8$^+$ or CD62L$^+$CD44$^-$CD4$^+$ T cells from WT mice were primed with anti-CD3/CD28 antibodies and TGF-β + IL-6 + IL-2 (murine Tc17) or TGF-β + IL-6 + IL-23 + IL-2 (murine Th17), in the presence of DMSO (Ctrl), 20 μM DMF or 20 μM DMF + 50 μM GSH (DMF + GSH) or 20 μM DMF + 1 mM NAC or ± 250 μM 2-DG or ± 10 nM Rotenone or ± 15 nM Oligomycin. **c, g, i, j** Flow cytometry of IL-17A in Tc17 **c, i, j** or Th17 cells **g** differentiated for 72 h with indicated treatment. **d** Ratio of reduced to oxidized glutathione (GSH/GSSG) contents in Tc17 cells differentiated for 2 h. **e, f** Flow cytometry of ROS levels in Tc17 or Th17 cells **f** differentiated for 2 h determined by CM-H$_2$DCFDA staining (fold of geometric mean fluorescence intensity (MFI), normalized to the corresponding control, which was arbitrarily set to 1. **h** Extracellular acidification rates (ECAR) and glycolytic capacity of Tc17 cells ± DMF after addition of 10 mM Glucose, 2.5 μM Oligomycin and 100 mM 2-DG. Bars show mean ± s.d. from seven **e**, or five **c, d, i**, or four to three **f, g, j** combined experiments, or one representative with 9 replicates of four experiments **h**; individual values are plotted. $P$-values in **a** from the same patient by two-tailed, paired $t$-test, while for the change in percentages CD8$^+$CD45RA$^-$IL-17$^+$ or CD4$^+$CD45RA$^-$IL-17$^+$ after DMF treatment by two-tailed, unpaired $t$-test (value between responders and non-responders), in **b** $P$ value and in **g, h** ***$p < 0.05$ or non-significant (ns), by two-tailed, unpaired $t$-test, **f** *$p < 0.05$ by two-tailed, unpaired $t$-test with Welch's correction, **c, i, j** ***$p < 0.001$, ****$p < 0.0001$ by one-way ANOVA followed by Tukey's Honestly Significant Difference (HSD) multiple comparison test, in **d, e** *$p < 0.05$, **$p < 0.01$, ***$p < 0.001$ by one-way Welch's ANOVA with Games–Howell multiple comparison test.

**DMF decreases *RORC*-to-*TBX21* ratio in memory CD8$^+$ T cells.** To identify the impact of DMF at a genome-wide level we performed RNA-Sequencing (RNA-Seq) of murine Tc17 cells treated with DMF alone or in combination with GSH. Among 281 transcripts highly significantly regulated by DMF (p adj < 0.01, log2FC ≥ 0.75), genes associated with type 17 T cells, including *Il17a*, *Il17f*, *Il21*, *Rorc*, and *Ccr6*, were downregulated, whereas the effector cytotoxic T lymphocyte (CTL) signature genes, *Ifng*, *Gzmb*, *Gzmc*, *Prf1*, and *Tbx21*[40], were upregulated (Fig. 2a). This was reflected by a decreased ratio of RORγt-to-T-BET at mRNA and protein levels (Fig. 2b, c). Both, positive and negative effects on gene transcription were reversed to a strong extent by the addition of GSH (Fig. 2a–c, Supplementary Fig. 2a), confirming a marked dependency on GSH-depletion. Gene set enrichment analysis (GSEA)[41] revealed that Tc17 signature genes were downregulated by DMF, while genes associated with effector CTL signature, were upregulated (Fig. 2d, e). Hence, DMF promotes a transcriptional shift of Tc17 cells towards a "CTL-like" transcriptional signature.

To elucidate whether the above described DMF effect also applied to human disease, we performed RNA-Seq of memory CD45RA$^-$CD8$^+$ T cells from PB of MS patients, which were not treated with DMF ("DMF-untreated", $n = 4$) or treated with DMF ("DMF-treated" $n = 4$) fulfilling NEDA-3 criteria after one year of therapy (Supplementary Fig. 2b, Supplementary Table 4). Principal component analysis (PCA) analysis of the top 20,000 genes clearly separated patients before and after DMF (Supplementary Fig. 2c). Upon DMF treatment 3840 transcripts were differentially expressed (DE) in human CD8$^+$ T cells ($p$ adj < 0.1), 965 transcripts of which were also differentially expressed in mouse Tc17 cells upon DMF treatment (Fig. 2f). Within the concordantly upregulated genes (Fig. 2g, upper right quadrant), we found transcripts associated with the effector CTL signature, *GZMB*, *IFNG*, *PRF1* and *TBX21*, whereas Tc17 signature genes *RORC*, *CCR6*, *IL23R*, *RORA* were downregulated accordingly (Fig. 2g lower-left, Supplementary Fig. 2d). Indeed, comparison of the top DE genes in the mouse with the human dataset revealed similar expression patterns for the majority of genes, including Tc17 and effector CTL signatures genes (Fig. 2h). Accordingly, similar to mouse data, the ratio of *RORC*-to-*TBX21*, was significantly reduced in DMF-treated human CD8$^+$CD45RA$^-$ T cells (Fig. 2i). This shift was likely regulated by ROS, as the ROS pathway was upregulated in memory CD8$^+$ T cells after one year of positive response to DMF as compared to therapy start (Fig. 2j and Supplementary Fig. 2e). Next, we compared memory CD8$^+$ T cells from MS patients to the IL-17$^+$CD8$^+$ and IL-17$^-$CD8$^+$ T cells from healthy individuals, which characteristic transcriptional profiles were recently determined[42]. Differentially expressed genes ($p$ adj < 0.0005) specific for IL-17$^+$CD8$^+$ T cell or IL-17$^-$CD8$^+$ T cell profile distinguished DMF-treated versus untreated MS patients. Interestingly, CD8$^+$ T cells from untreated patients exhibited more similarity to IL-17$^+$CD8$^+$ T cells than cells from DMF-treated patients, which in turn were more similar to IL-17$^-$CD8$^+$ T cells, corroborating the idea on DMF-mediated diversion of Tc17 towards a "CTL-like" transcriptional signature (Fig. 2k, l, Supplementary Fig. 2f, g).

**PI3K-AKT-T-BET axis suppresses IL-17 and RORγt in Tc17 cells.** Analysis of pathways involved in a positive response to DMF therapy (defined as fulfillment of NEDA-3 criteria) in memory CD8$^+$ T cells from MS patients revealed a significant enrichment for genes associated with the PI3K-AKT-mTOR-pathway (Fig. 3a and Supplementary Fig. 3a) (GSEA, MSigDB, hallmark dataset). Indeed, inhibition of PI3K activity by the inhibitor Ly294002 resulted in partial restoring of IL-17 production in DMF-treated murine Tc17 cells (Fig. 3b), suggesting that enhanced PI3K-signaling in DMF-treated Tc17 cells contributed to IL-17 suppression. Furthermore, downstream of PI3K, phosphorylation of AKT[43] at S473 as well as at T308 (Fig. 3c, d) was enhanced by DMF, GSH-dependently.

AKT phosphorylates the transcription factor FOXO1 to inactivate its transcriptional activity[44]. Consistent with this notion, DMF increased FOXO1/3a phosphorylation, accompanied by the downregulation of FOXO1 targets (Fig. 3e, f). FOXO1 is a suppressor of T-BET[44,45] and T-BET was upregulated in DMF-treated Tc17 cells (Fig. 2c). We therefore speculated that DMF-mediated suppression of IL-17 was dependent on AKT-dependent FOXO1 inactivation, leading to upregulation of T-BET. To test this hypothesis, we compared the susceptibility of T-BET-deficient (*Tbx21$^{-/-}$*) and wildtype (WT) Tc17 cells to treatment with DMF and a selective AKT-1/2 inhibitor (AKTi)[46]. AKTi boosted IL-17 production significantly in DMF-treated WT Tc17 cells (Fig. 3g, upper). In contrast, the impaired IL-17 production in DMF-treated *Tbx21$^{-/-}$* Tc17 cells was refractory to AKTi treatment (Fig. 3g, lower), indicating that T-BET is required for AKT-mediated IL-17 suppression. Furthermore, DMF failed to inhibit RORγt and was less effective in suppressing IL-17 in *Tbx21$^{-/-}$* as compared to WT cells (Fig. 3h, i). Contrary to Tc17 cells, AKTi in combination with DMF inhibited IL-17 production in pathogenic Th17 cells (Supplementary Fig. 3b). Another target of PI3K-AKT signaling is mTORC1[43]. DMF treatment resulted in an enrichment of mTOR-associated transcripts in CD8$^+$ T cells from MS patients and in murine

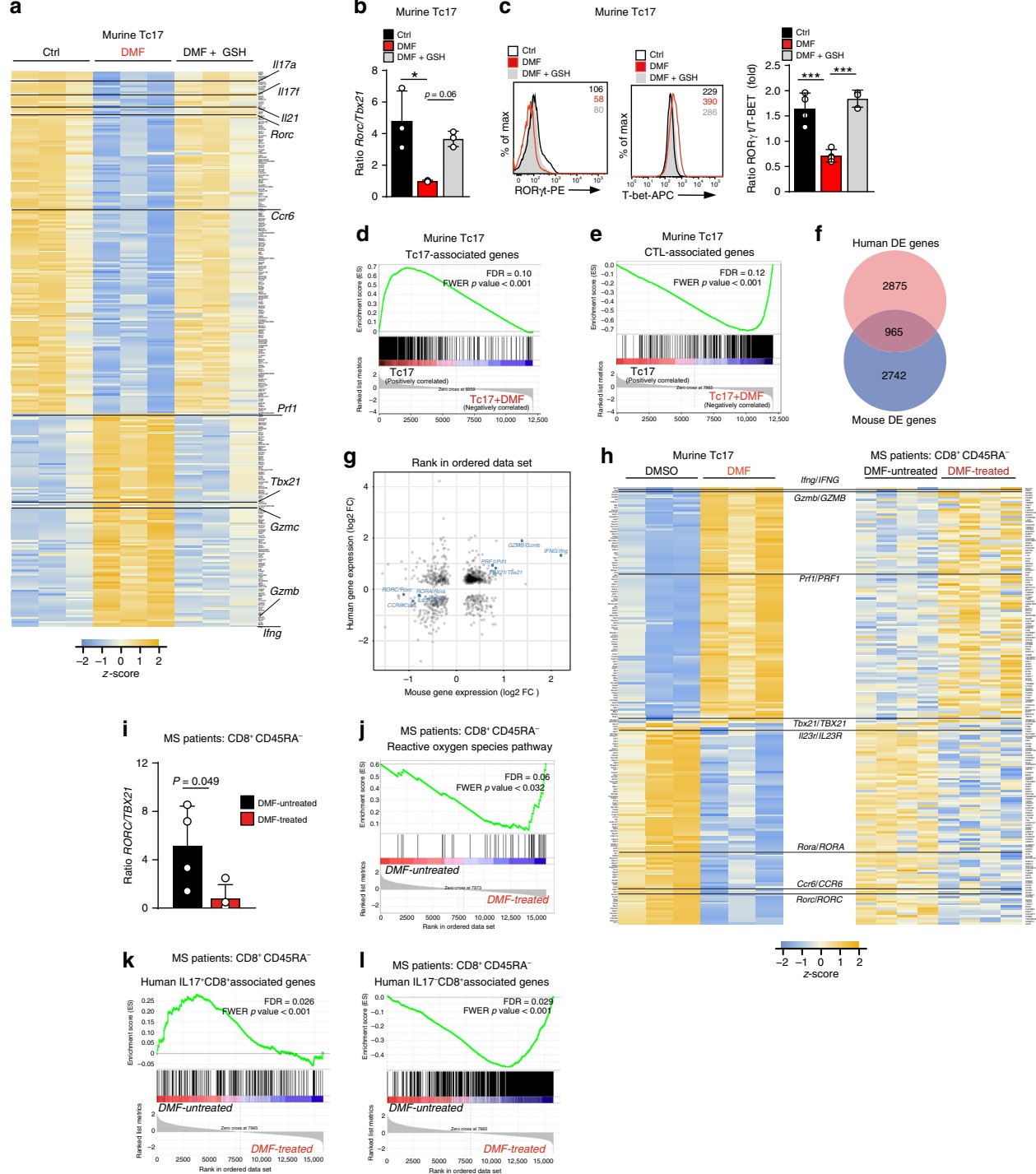

Tc17 cells (Supplementary Fig. 3c, d). Furthermore, DMF enhanced the mTORC1-dependent phosphorylation of the ribosomal protein S6 at S235/236 in murine Tc17 cells (Supplementary Fig. 3e). In line with published data[39], inhibition of mTOR signaling by rapamycin blocked IL-17 production in pathogenic Th17 cells (Supplementary Fig. 3f), in contrast to Tc17 cells in which rapamycin did not impact IL-17 (Supplementary Fig. 3g).

In summary, these data reveal a crucial role of the PI3K-AKT-FOXO1 pathway in the DMF-mediated IL-17 suppression as well as in the shift in RORγt-to-T-BET ratio and indicate differential roles of AKT and mTOR signaling in IL-17 regulation by Tc17 and Th17 cells.

**STAT5 contributes to inhibition of IL-17 in Tc17 cells**. Deeper analysis of the transcriptional profiling data revealed an enrichment of IL-2-STAT5 signaling-associated genes in DMF-treated murine Tc17 cells, which was GSH-dependent (Fig. 4a and Supplementary Fig. 4a). In line with a role for this pathway, DMF inhibited IL-17 production in Tc17 cells only in the presence of IL-2, in a dose-dependent manner (Fig. 4b, c). However, neither GSH depletion nor ROS upregulation were dependent on the presence of IL-2 (Fig. 4d, e). Thus, DMF-induced GSH depletion, which correlates with ROS upregulation suppressed IL-17 production by enhancing IL-2 signaling.

IL-2 signaling involves several pathways, including activation of PI3K-AKT and activation of STAT5[47], which inhibits IL-17

**Fig. 2 Diversion of Tc17 cells towards a "CTL-like" genetic signature upon DMF treatment. a** RNA-Seq based heatmap of 281 differentially expressed (DE) genes (p adjusted [p adj] < 0.01 and log2FC ≥ 0.75) between Ctrl ± DMF or DMF + GSH-treated murine Tc17 cells differentiated for 48 h (n = 3). Highlighted are genes associated with type 17 or CTL signature. **b** Expression-ratio of *Rorc*-to-*Tbx21* calculated from RNA-Seq from **a**, normalized to the DMF values, which were arbitrarily set to 1. **c** Flow cytometry of RORγt or T-BET in murine Tc17 cells differentiated for 72 h, to the right, ratio of RORγt-to-T-BET calculated from fold MFI. **d**, **e** GSEA of genes associated with Tc17 **d** or CTL **e** phenotype as defined by GSE110346 in Tc17 cells from **a**. **f** Venn diagram of DMF-dependent DE genes in murine Tc17 (dataset from **a**) and human CD8+CD45RA− T cells from matched groups of MS patients "DMF untreated" (n = 4) and "DMF treated" (n = 4), who fulfilled NEDA-3 criteria after 12–13 months of DMF therapy (Supplementary Table 4 and Supplementary Fig. 2b) based on RNA-Seq (p adj < 0.1). **g** Scatter plot of overlapping gene regulation in murine Tc17 and human CD8+CD45RA− T cells datasets from **a** and **f**, respectively (p adj < 0.1). Highlighted are concordantly expressed genes associated with Tc17 or CTL phenotype. **h** Heatmap of top transcripts with correlating expression in murine Tc17 and human CD8+CD45RA− T cell-datasets from **a**, **f**, respectively. DE mouse Tc17 transcripts (p adj < 0.01, log2Fc ≥ 0.6), and corresponding 182 human transcripts with GSEA core enrichment were selected. Highlighted are genes associated with Tc17 and CTL phenotype. **i** Relative expression of *RORC*-to-*TBX21* calculated from RNA-seq from **f**. **j** GSEA of genes associated with ROS-signaling in human CD8+CD45RA− T cells from **f** based on MSigDBv6·1. **k**, **l** GSEA of genes associated with IL17+CD8+ **k** or IL17−CD8+ **l** profiles in CD8+CD45RA− T cells from **f** based on published raw data (RNA-Seq GSE96741)[42]. Bars show mean ± s.d. from four to three **b**, **c**, **i** combined experiments; individual values are plotted. In **b**, **c** *p < 0.05, ***p < 0.001 evaluated by one-way ANOVA followed by Tukey's HSD multiple comparison test, in **i** p-value by the two-tailed, unpaired t-test.

production in Th17 cells[48]. DMF upregulated IL-2-mediated STAT5 phosphorylation GSH-dependently (Fig. 4f), and retro-viral overexpression of constitutively active STAT5 revealed that STAT5 suppressed IL-17 production in Tc17 cells (Fig. 4g and Supplementary Fig. 4b, c). Conversely, inhibition of STAT5 function by a pharmacological inhibitor partially restored IL-17 production (Fig. 4h). These data indicate that DMF acts on IL-2 signaling, involving both PI3K-AKT-T-BET and STAT5 path-ways to suppress IL-17 production in Tc17 cells (Fig. 4i).

**DMF drives epigenetic remodeling at the *Il17* locus.** Histone modifications at gene regulatory elements undergo dynamic changes that correlate with gene expression profiles[49]. Since DMF limited IL-17 in Tc17 cells, we speculated on accompanying changes in the epigenetic landscape. Indeed, DMF suppressed permissive H4Ac as well as H3K27Ac on *Il17* promoter and enhancer-5 (Fig. 5a), partially in GSH-dependent manner. In contrast, neither the global histone acetylation nor the *Il10* pro-moter were affected, indicating a specific suppression of the *Il17* locus (Fig. 5b and Supplementary Fig. 5a). Since IL-2 signaling depends on histone deacetylases (HDACs)[50], and DMF inhibited IL-17 in IL-2-dependent manner, we assumed an influence of DMF-triggered pathway on HDACs. The inhibitor of class I and II of mammalian HDACs, trichostatin A (TSA), neutralized the inhibitory effect of DMF on IL-17 production to some extent (Supplementary Fig. 5b), indicating a partial involvement of type I or II HDACs in DMF-driven suppression of histone acetylation on the *Il17* locus.

Furthermore, permissive H3K4me3 on the *Il17a* promoter, as well as to some extent on the *Il17* enhancer-5 was suppressed by DMF treatment (Fig. 5c), while the H3K4me3 on *rpl32* promoter or repressive H3K27me3 were not significantly altered (Fig. 5d and Supplementary Fig. 5c–e). Thus, DMF-signaling leads to suppression of permissive histone modifications on the *Il17* locus.

**DMF impairs the pathogenicity of Tc17 cells in EAE.** To test the impact of DMF in vivo, we induced EAE in WT mice and treated them with DMF by oral gavage starting from disease onset (day 8, therapeutic setting) or in drinking water staring ten days before immunization (preventive setting) (Supplementary Fig. 6a, b). Therapeutic DMF application significantly reduced EAE severity, reflected by reduced T cell and CD8+ T cell numbers, as well as by decreased frequency of Tc17 cells in the CNS (Fig. 6a–d). Con-sistent with the literature[26,29], preventive DMF treatment likewise reduced EAE severity (Supplementary Fig. 6c). This was accom-panied by significantly decreased percentages of Tc17 cells in CNS (Supplementary Fig. 6d), in accordance with the data obtained in the therapeutic setting (Fig. 6d). Thus, therapeutic and preventive

DMF application caused EAE amelioration accompanied by a reduction in the Tc17 cell abundance. To investigate DMF effects specifically on Tc17 cells, we applied an adoptive transfer EAE model involving the cooperating Tc17 and Th17 cells, in which Tc17 cells via IL-17A provide "reverse help" for CNS pathogeni-city of Th17 cells[12]. To this end, we transferred sub-pathogenic numbers of 2D2 CD4+ T cells, which are transgenic for myelin oligodendrocyte glycoprotein (MOG)-specific Vβ11+/Vα3.2+ TCR together with congenic polyclonal Tc17 cells into *Irf4*−/− mice, which are resistant to EAE[12,51]. Tc17 cells were treated with DMF or control during in vitro differentiation. Transfer of 2D2 cells alone did not evoke disease, while the combination of 2D2 cells with Tc17 cells caused early onset and severe disease course (Fig. 6e). This was accompanied by a prominent T cell infiltration into the CNS, including endogenous and transferred CD8+ T cells (Fig. 6f, g). Transferred Tc17 cells were detectable in draining LNs and correlated with CNS-infiltration of CD4+ T cells producing IL-17 at higher proportions as compared to transferred 2D2 cells alone (Fig. 6h–k). In contrast, co-transfer of 2D2 cells together with DMF-treated Tc17 cells failed to evoke severe EAE and the total T cell and CD8+ T cell infiltration into the CNS was strongly reduced (Fig. 6e–g). Accordingly, transferred DMF-treated Tc17 cells produced less IL-17 in draining LNs and failed to upregulate IL-17 production by CNS infiltrating CD4+ T cells (Fig. 6h–k). Hence, DMF treatment caused a loss of the Tc17-dependent Th17 pathogenicity and conferred a stable "low IL-17" phenotype to Tc17 cells, suggesting a mechanism for amelioration of auto-immunity in CNS upon DMF treatment.

## Discussion

Over the past decades, several MS-modifying drugs have been approved for treatment. This includes orally applied DMF, which alleviates disability progression and has good safety and toler-ability[25]. DMF reduced the annual relapse rate by 53% and relative risk reduction of disability progression by 38%[36], indi-cating its efficiency. However, a considerable proportion of patients do not respond to DMF therapy, indicating a need for stratification of patients. Although DMF has been shown to act on different cell types including microglia, neurons, dendritic cells, macrophages and to some extent CD4+ T cells, it is still not known how this drug reduces overall disease activity[25]. Therefore, a close mechanistic understanding of the influence of DMF on target cells can contribute to a development of markers and an improvement of its clinical efficacy.

IL-17A plays an important role in the autoimmune-patho-genesis, since (i) increased numbers of IL-17+ CD4+ and CD8+ T cells are detectable in active as compared to inactive areas of MS lesions[15], (ii) genetic risk factors related to IL-23-IL-17 axis

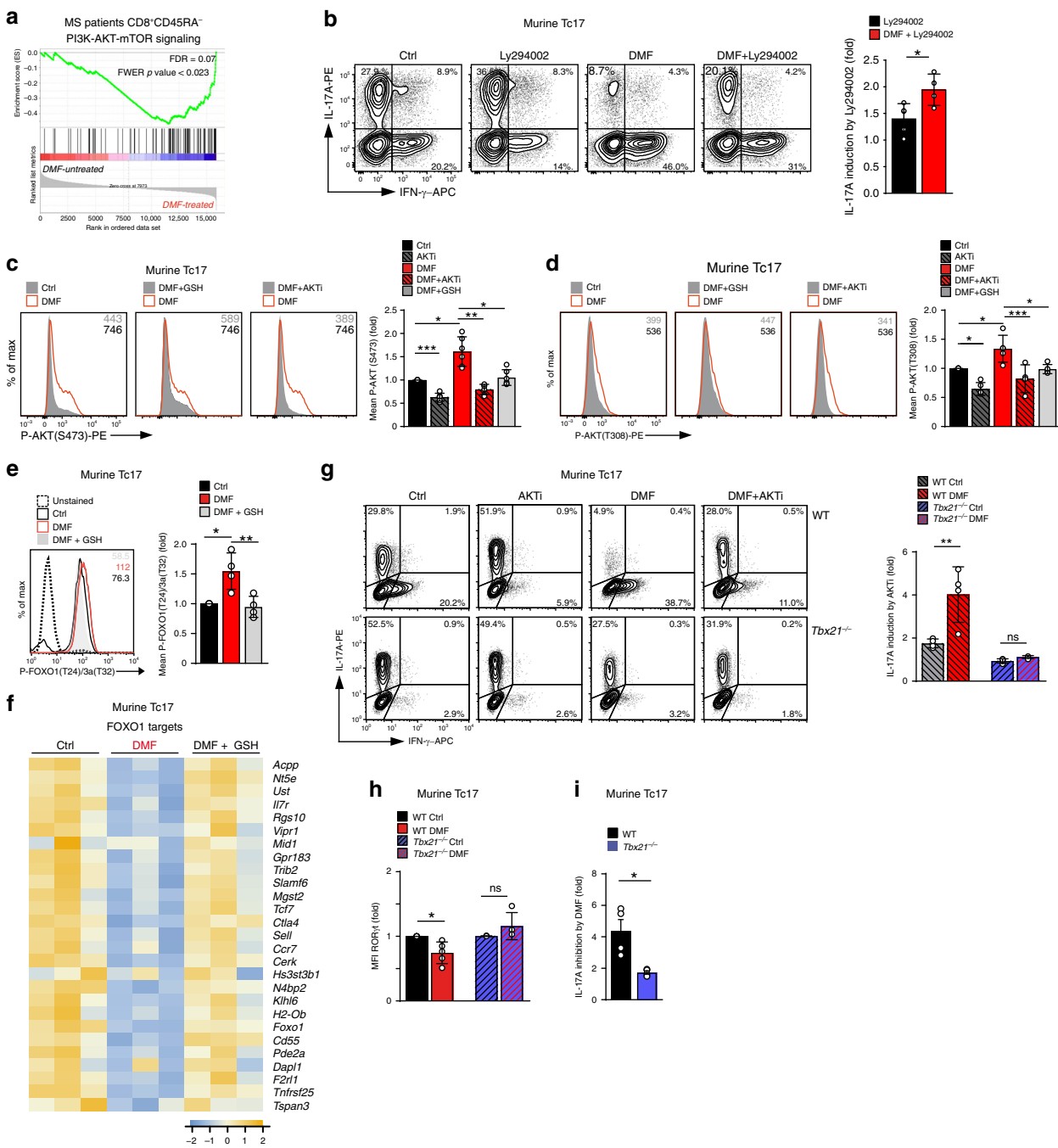

**Fig. 3 DMF enhances PI3K-AKT-T-BET-signaling to diminish IL-17 and RORγt in Tc17 cells. a** GSEA examining the enrichment of genes associated with PI3K-AKT-mTOR-signaling in human CD8+CD45RA− T cells upon stable response to DMF therapy based on MSigDBv6.1 (dataset from Fig. 2f). **b** Flow cytometry of IL-17A in murine Tc17 cells differentiated for 72 h ± DMF, ±1 μM Ly294002 (fold IL-17A induction by Ly294002, a PI3K inhibitor). **c**, **d** Flow cytometry of P-AKT(S473) **c** and P-AKT(T308) **d** in murine Tc17 cells differentiated for 48 h ± DMF, DMF + GSH, ±1 μM AKTi (AKT-1/2 inhibitor) or DMF + AKTi. **e** Flow cytometry of P-FOXO1(T24)/FOXO3a(T32) in Tc17 cells differentiated for 48 h ± DMF or DMF + GSH. Bars to the right in **c**, **d**, and **e** show fold MFI normalized to the respective control, which was arbitrarily set to 1. **f** Heatmap of color-coded z-scores from the rlog transformed, batch-corrected FOXO1 target genes according to Michelini et al[44] in Tc17 cells, (dataset from Fig. 2a). **g** Flow cytometry for IL-17A and IFN-γ in WT and Tbx21−/− Tc17 cells differentiated for 72 h with indicated treatment (fold IL-17A induction by AKTi). **h** Flow cytometry for RORγt in WT and Tbx21−/− Tc17 cells differentiated for 72 h with indicated treatment. Bars show fold RORγt expression (MFI normalized to respective control, which was arbitrarily set to 1). **i** Flow cytometry for IL-17A in WT and Tbx21−/− Tc17 cells differentiated for 72 h (fold IL-17A inhibition by DMF). Bars show mean ± s.d. from six **c**, five to four **b**, **d**, **e** and **g**–**i** combined experiments; individual values are plotted. In **b**, **g**, **i** *$p < 0.05$, **$p < 0.01$ evaluated by two-tailed, unpaired *t*-test, in **h** *$p < 0.05$ by two-tailed unpaired *t*-test with Welch's correction, in (d and e) *$p < 0.05$, **$p < 0.01$, ***$p < 0.001$ by one-way ANOVA followed by Tukey's HSD multiple comparison test, in **c** *$p < 0.05$, **$p < 0.01$, ***$p < 0.001$ by one-way Welch's ANOVA with Games-Howell multiple comparison test.

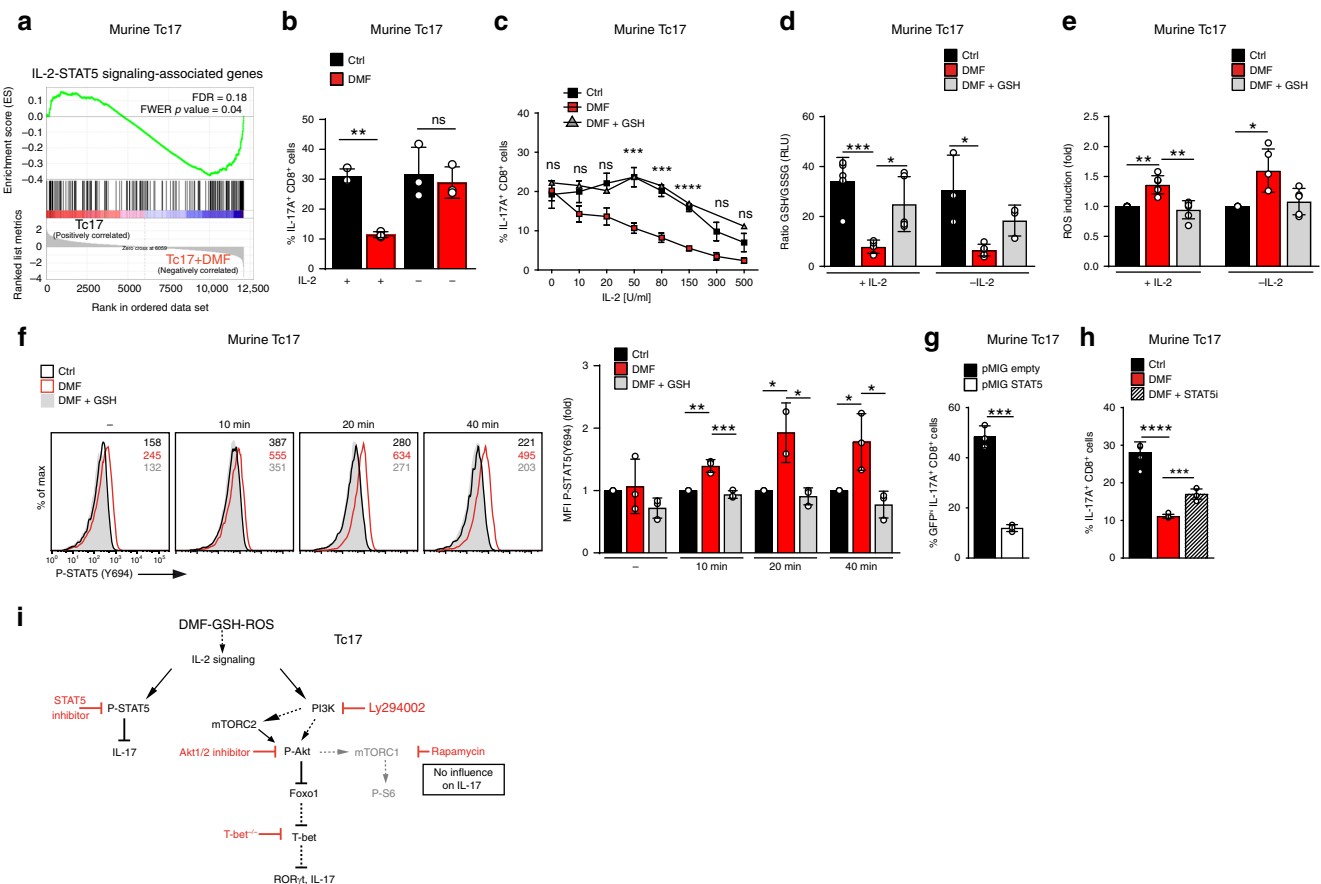

**Fig. 4 DMF interferes with IL-2-STAT5 signaling to limit IL-17 expression in Tc17 cells. a** GSEA of genes associated with IL-2-STAT5-signaling in murine Tc17 cells ± DMF (data set from Fig. 2a) based on MSigDBv6.1. **b, c** Flow cytometry of IL-17A in murine Tc17 cells differentiated for 72 h ± IL-2, ± DMF or DMF + GSH. Significances in graph are shown for Ctrl vs DMF (mean ± s.d.). **d** GSH/GSSG ratio in murine Tc17 cells differentiated for 2 h ± rhIL-2, ± DMF or DMF + GSH. **e** Flow cytometry of ROS levels determined by CM-H$_2$DCFDA staining in murine Tc17 cells differentiated for 2 h ± IL-2 or DMF + GSH (fold MFI normalized to the respective control, which was arbitrarily set to 1). **f** Flow cytometry of phospho(P)-STAT5(Y694) in murine Tc17 cells differentiated for 48 h, rested, then activated with IL-2 ± DMF or DMF + GSH for the indicated time (fold MFI normalized to control, which was arbitrarily set to 1). **g, h** Flow cytometry of IL-17A in murine Tc17 cells after retroviral transduction with constitutive-active P-STAT5 (pMIG STAT5) or GFP vector alone (pMIG empty) **g** or in Tc17 cells after 72 h of differentiation ± DMF, ± 35 µM STAT5 inhibitor (STAT5i) **h**. **i** Schematic influence of DMF-GSH-ROS on IL-2 signaling, leading to IL-17 and RORγt suppression in Tc17 cells. Modified from[43]. Bars show mean ± s.d. from seven to five **e**, six **h**, or five to three **d** or three **b, c, f, g** combined experiments; individual values are plotted. In **g** ***$p < 0.001$, evaluated by two-tailed, unpaired $t$-test, in **b–d**, **f**, **h** *$p < 0.05$, **$p < 0.01$, ***$p < 0.001$, by one-way ANOVA followed by Tukey's HSD multiple comparison test, in **e** *$p < 0.05$, **$p < 0.01$ by one-way Welch's ANOVA with Games-Howell multiple comparison test.

associate with MS[3,5], (iii) increased IL-17A mRNA levels are detectable in MS[52] and (iv) IL-17 contributes to disruption of blood–brain–barrier tight junctions[53]. Furthermore, the therapeutic targeting of IL-17A by the fully humanized antibody secukinumab is successful in psoriasis[54], rheumatoid arthritis[55] and ankylosing spondomyelitis[56], as well as there are promising results in a proof-of-concept study in MS[57]. Here, we demonstrate that a positive response to DMF therapy associates with a reduction in Tc17, in contrast to Th17 cells, in MS. Furthermore, in patients with a positive response to the therapy, DMF modified Tc17 transcriptional profile towards a "CTL-like" signature. Indeed, DMF inhibited IL-17 production in human Tc17 cultured in vitro, indicating its direct effect. Similar pattern of response to DMF was observed in murine Tc17 cells, revealing a comparable regulation across species and thus allowing mechanistic and functional experiments. In the mouse model, oral DMF treatment in therapeutic as well as in preventive setting ameliorated clinical signs of disease and suppressed frequency of IL-17-producing CD8$^+$ T cells in CNS, consistent with the results obtained from PB of MS patients. In the adoptive transfer model, the

amelioration of the disease by DMF was caused by a stable suppression of Tc17 cells and thereby loss of their co-pathogenic function resulting in the reduced frequency of Th17 cells in the CNS. This is consistent with previous reports showing reduced frequencies of Th17 cells in DMF-treated mice[26,28]. Immune modulatory effects of DMF also include influence on IFN-γ production by CD4$^+$ T cells[26,28], as well as on the phenotype of dendritic cells, monocytes[26,29,32] and metabolism of macrophages[28], which likewise contribute to the therapeutic effect. Considering multiple mechanisms driven by DMF and an extensive heterogeneity in the disease course resulting from distinct effector mechanisms underlying MS[1], we believe that our findings and conclusions apply to a subset of patients, in which Tc17 cells are involved in the disease pathogenesis. This idea is supported by our finding that the mean frequency of Tc17 cells before DMF therapy was significantly higher in responders as compared to non-responders however, further studies should prove this concept.

In MS patients the majority of IL-17-producing CD8$^+$ T cells expresses the molecules CD161 and CCR6, as well as TCRVα7.2,

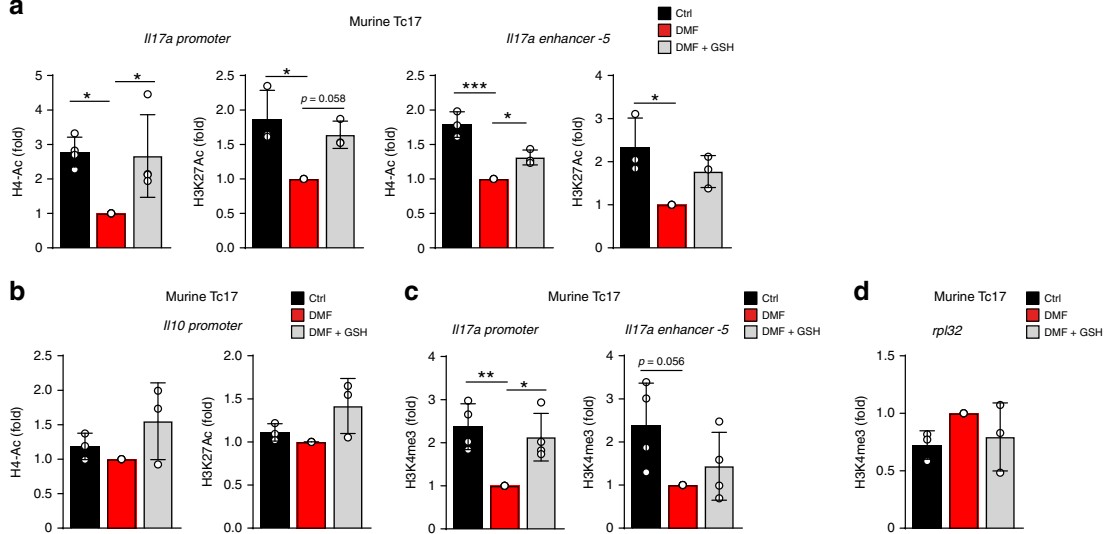

**Fig. 5 Suppression of histone modifications on *Il17a* promoter by DMF is GSH-dependent. a** ChIP assays for H4Ac and H3K27Ac at the *Il17a* promoter, *Il17* enhancer-5 in murine Tc17 cells differentiated for 72 h ± DMF or DMF + GSH. **b** ChIP assays for H4Ac and H3K27Ac at the *Il10* promoter in murine Tc17 cells differentiated for 72 h ± DMF or DMF + GSH. **c** ChIP assays for H3K4me3 at the *Il17* promoter, *Il17* enhancer-5 in murine Tc17 cells differentiated for 72 h ± DMF or DMF + GSH. **d** ChIP assays for H3K4me3 at the *Rpl32* promoter in murine Tc17 cells differentiated for 72 h ± DMF or DMF + GSH. Bars show mean ± s.d. of fold change normalized to DMF treatment, which was arbitrarily set to 1. Data from three **a–d** combined experiments; individual values are plotted. In **a–d**, *$p < 0.05$, **$p < 0.01$, ***$p < 0.001$, evaluated by one-way ANOVA followed by Tukey's HSD multiple comparison test.

characterizing them as MAIT cells[16,17,19]. Functionally, these cells seem to have pathogenic relevance since pediatric MS patients harbored more IL-17-producing MAIT cells in PB, as compared to healthy controls or children with monophasic inflammatory CNS disorder[18]. Furthermore, increased IL-17 production by MAIT cells[20] as well as their enhanced accumulation in brain lesions[19] was detectable in MS. Considering these reports, it is possible, that the DMF-mediated reduction of IL-17 producing CD8+ T cells in MS patients, which we herein described, may also affect CD161high CD8+ MAIT cells. As MAIT cells were not affected by IFN-β therapy, but strongly reduced by high dose of cyclophosphamide in combination with alemtuzumab treatment followed by autologous stem cell transplantation in MS patients[17], one can speculate that besides the non-myeloablative depletion, MAIT cells might be susceptible to ROS upregulated by DMF-mediated glutathione depletion. Therefore, it will be of interest for future studies to compare in detail the susceptibility to DMF of conventional versus CD161high CD8+ MAIT Tc17 cells in patients, to define the main target population within IL-17-producing CD8+ T cells and to closely characterize the therapy responder group.

The idea of DMF-mediated diversion of Tc17 profile towards a "CTL-like" genetic signature is based on the comparison of memory CD8+ T cells from untreated versus treated MS patients with IL-17+CD8+ versus IL-17−CD8+ T cell from healthy individuals, which profiles were recently published[42], as well as with our datasets on gene expression in mouse effector Tc17. Importantly, for the gene expression profiles of human IL-17+CD8+ and IL-17−CD8+ T cells, MAIT and γδT cells were excluded[42], suggesting that the DMF-mediated diversion from Tc17-like towards "CTL-like" profile relates rather to "classical" Tc17 cells. The diversion was accompanied by a decreased ratio of RORγt-to-T-BET, transcriptional regulators governing the development of type 17 or CTL, respectively. This indicates that DMF not only simply inhibited IL-17, but suppressed the transcriptional program governing Tc17 differentiation in favor of T-BET and CTL-associated genes. DMF-mediated upregulation of T-BET crucially contributed to this effect, as T-BET opposes Tc17

differentiation[22,24] and RORγt[58] expression. Our work further unraveled the molecular pathway of DMF mediated T-BET upregulation, which included enhanced AKT activation, accompanied by FOXO1 phosphorylation and inactivation, leading to upregulation of T-BET. Accordingly, T-BET deficiency in Tc17 cells led to decreased response to DMF with respect of IL-17 production and RORγt downregulation. These data reveal T-BET as an important DMF target in Tc17 cells, suppressing IL-17 production and RORγt expression. In line with our data, a previous report demonstrated significantly reduced *TBX21* expression in MS patients compared to healthy controls[59], pointing to an association of low *TBX21* levels with the disease. Therefore, we surmise that upregulation of *TBX21* with concomitant downregulation of *RORC* in CD8+ T cells by DMF contributes to its therapeutic efficacy in MS. Further studies might support this speculation.

In addition to the PI3K-AKT-T-BET pathway, enhanced STAT5 activity contributed to DMF-mediated suppression of IL-17, in which STAT5 limited IL-17 production in Tc17 cells in a similar manner as described for Th17 cells[48]. The STAT5 or PI3K-AKT activity-enhancing function by glutathione-depletion has not been described for T cells so far, but upregulated ROS enhanced STAT5-phosphorylation and PI3K-AKT-activity in hepatocytes in the context of obesity and type II diabetes[60] or cancer cells[61,62]. Considering that upregulated ROS can inhibit phosphatase activity, as demonstrated for the protein-tyrosine phosphatase PTPN2 in hepatocytes[60] and for PTEN in macrophages[62], it is conceivable that the increased phosphorylation of PI3K-AKT-FOXO1 and STAT5 is caused by a similar mechanism in Tc17 cells. DMF-signaling led to suppression of the permissive histone state on *Il17* locus, ultimately restricting IL-17. Notably, and in contrast to STAT5, AKT-mediated suppression of IL-17 production took place in Tc17 but not in pathogenic Th17 cells. This indicates common (STAT5) and differential signaling pathways (AKT) regulating IL-17 in Tc17 and Th17 cells. Furthermore, Tc17 and Th17 cells seem to differ in respect to utilized energy supply. Whereas Th17 cells strongly depend on glycolysis[39], Tc17 cells were refractory to 2-DG treatment, instead they seem to rely

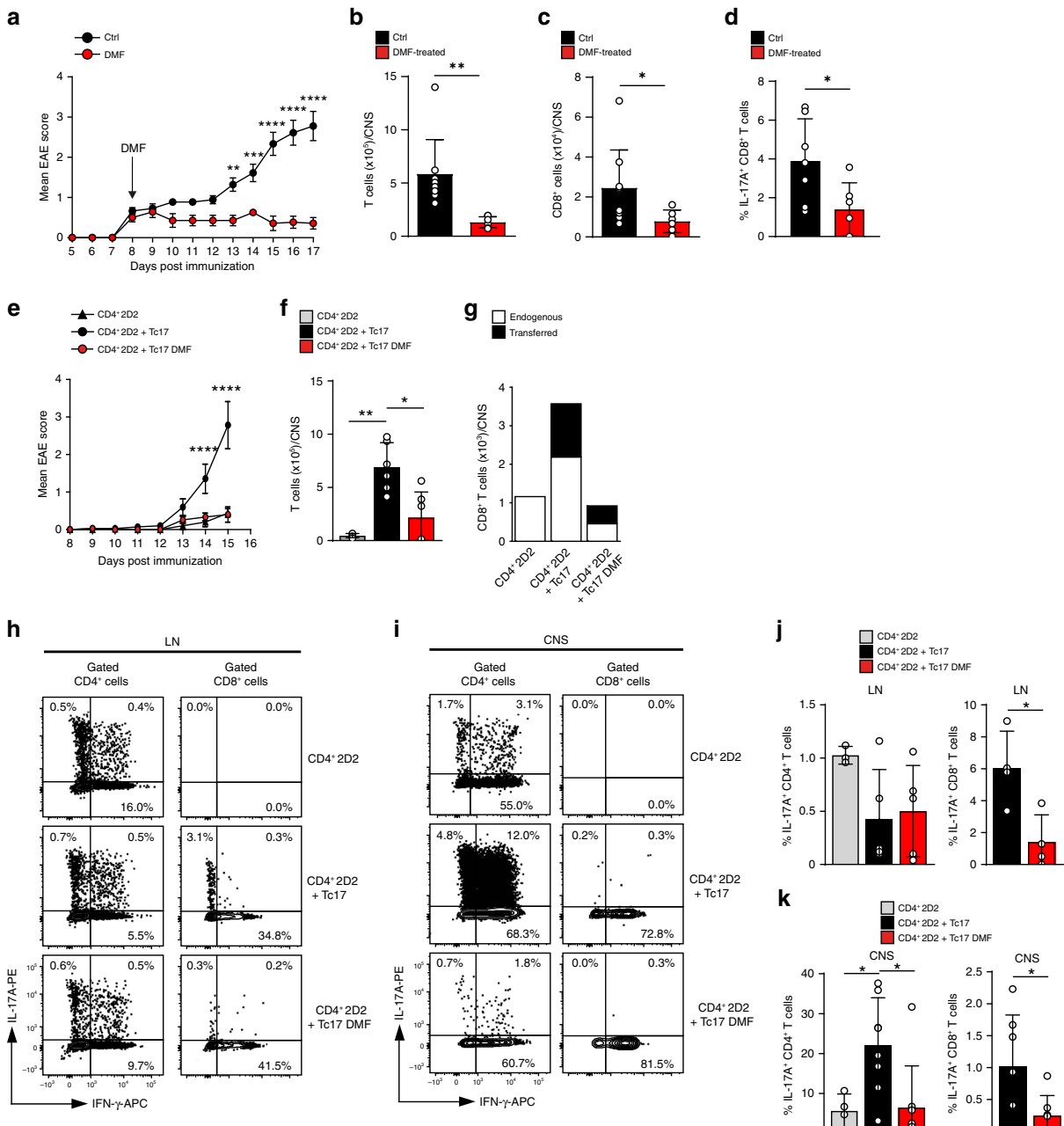

**Fig. 6 DMF restricts CNS autoimmunity by limiting the co-pathogenic function of Tc17 cells. a** Mean clinical scores ( ± s.d.) of MOG$_{35-55}$ immunized wild-type (WT) mice ($n = 7$) with therapeutic DMF application, starting from day 8, indicated by arrow. **b** T cell numbers (mean ± s.d.; $n = 7$) in the CNS of WT mice ± DMF treatment. **c** CD8$^+$ T cell numbers (mean ± s.d.; $n = 7$) in the CNS of WT mice ± DMF treatment. **d** Percentages of IL-17A$^+$CD8$^+$ T cells (mean ± s.d.; $n = 7$) in the CNS of WT mice ± DMF treatment. **e** Mean clinical scores ( ± s.d.) of MOG$_{35-55}$ immunized *Irf4*$^{-/-}$ ($n = 7$) mice receiving 10$^3$ CD4$^+$2D2 T cells alone or together with 2.5 × 10$^6$ in vitro differentiated Tc17 ± DMF. Significances in graph are shown for 2D2 + Tc17 vs 2D2 + Tc17 DMF. **f** T cell numbers (mean ± s.d., $n = 4$-6) in CNS of *Irf4*$^{-/-}$ mice after CD4$^+$2D2 T and Tc17 ± DMF co-transfer. **g** Endogenous to transferred CD8$^+$ T-cell ratio (mean, $n = 5$) in CNS of *Irf4*$^{-/-}$ mice after CD4$^+$2D2 T and Tc17 ± DMF co-transfer. **h, i** Flow cytometry of IL-17A and IFN-γ in gated CD4$^+$ and CD8$^+$ T cells in LN **h** or CNS **i** of *Irf4*$^{-/-}$ mice after CD4$^+$2D2 T and Tc17 ± DMF co-transfer. **j, k** Percentages of IL-17A$^+$CD4$^+$ or IL-17A$^+$CD8$^+$ T cells (mean ± s.d.) in LN ($n = 4$-5) **j** and CNS ($n = 7$-8) **k** of *Irf4*$^{-/-}$ mice; **b, c, d, f, j, k** individual values are plotted. In **a, e** **$p < 0.01$, ***$p < 0.001$, ****$p < 0.0001$ evaluated by two-way ANOVA with Bonferroni post-hoc test, in (**d, j,** for CD8$^+$ T cells) *$p < 0.05$ by two-tailed, unpaired $t$-test, in (**b, c, k** for CD8$^+$ T cells) *$p < 0.05$, **$p < 0.01$ by two-tailed, unpaired $t$-test with Welch's correction, in (**j, k** for CD4$^+$ T cells) *$p < 0.05$, **$p < 0.01$ by one-way ANOVA followed by Dunnett multiple comparison test, in **f** *$p < 0.05$, **$p < 0.01$ by one-way Welch's ANOVA with Games–Howell multiple comparison test.

on OXPHOS as examined by sensitivity to inhibitors rotenone and oligomycin. Hence, our data indicate cell-type-specific signaling pathways controlling IL-17 production, which could explain particular responsivity of Tc17 to DMF treatment. This is in agreement with a recent publication demonstrating specific

transcriptional programs for mouse and human Tc17 and Th17 cells[42].

Overall, we provide mouse and human data that support a concept for DMF mode of action in MS that may endorse development of drugs targeting the delineated signaling pathways.

The described mechanisms can be extended to other IL-17-mediated diseases, where CD8$^+$ T cells are a cellular source, including psoriasis or tumorigenesis to develop comprehensive therapy strategies.

## Methods

**Study design and participants.** Patients visiting the Center of Neuroimmunology of the University of Marburg and at the Department of Neurology at the University of Mainz between 2014–2017 were offered standard immunotherapies according to national treatment guidelines and followed longitudinally in an observational cohort. Key eligibility criteria included age of 18 to 57 years, a diagnosis of MS (according to the 2010 McDonald criteria[63]), an Expanded Disability Status Scale (EDSS) score of 0 to 4.0 at the start of DMF therapy (score range from 0 to 10.0 with higher scores indicating a greater degree of disability), maximum one prior short-term therapy with IFN-β or glatiramer acetate to minimize effects on immune system and the ability to give informed consent. The key exclusion criteria were a diagnosis of primary progressive multiple sclerosis, prior therapies with monoclonal antibodies or immunosuppressive medications, disease duration of more than 2.5 year since diagnosis. Blood was collected immediately prior to treatment initiation and approx. one year later. Patients were followed longitudinally and stratified into two groups according to treatment response (Tecfidera, 240 mg per os twice a day) defined as NEDA-3 positive or negative status under an appropriate treatment period with DMF (re-baselining 4 months after treatment start). NEDA-3 is a composite score defined as: (i) no relapses, (ii) no sustained disability progression measured with EDSS and (iii) no new/enlarging T2-weighted lesions in magnetic resonance imaging (MRI)[37]. In total, 65 patients were included in the study cohort, compromising 40 responders and 25 non-responders. 3 non-responders were excluded as therapy was escalated due to disease activity after 4 months after treatment initiation. In agreement with DMF clinical efficacy, more patients were included into the responder group ($n = 40$). To exclude a selection bias and to obtain comparable group sizes, propensity score matching 1:1 to responders and to non-responders based on age, gender, MS duration since diagnosis and EDSS before treatment was performed. If multiple qualified patients were available for matching, random selection was employed. Selected ($n = 21$) and excluded ($n = 19$) responders did not differ concerning above mentioned parameters. From the measured 21 DMF-responder samples, 3 samples were excluded, while from 22 non-responder samples, 4 samples were excluded, all because of low cellularity within the primary sample (Supplementary Fig. 7).

Since NEDA-3 reflects therapeutic efficacy, patients, who were NEDA-3 positive were termed responders ($n = 18$), whereas NEDA-3 negative patients were termed non-responders ($n = 18$), (Supplementary Tables 1–3). All responders were not treated with any disease modifying therapy before initiation of DMF therapy ("treatment naïve") except of a standard of care short-term high dose steroid treatment (1000 mg methylprednisolone for 5 consecutive days), leading to symptom recovery. Blood was drawn 8 weeks after discontinuing from steroid treatment as a baseline and then after 12 months of DMF treatment, a time point of a clear clinical and immunological response to the drug[64,65]. The non-responder group comprised of 10 treatment naïve MS patients and 8 patients with previous therapy: 6 patients were treated with IFN-β-1a (44 μg s.c., 3 times a week) for 0.5–2 years, whereas 2 patients received glatiramer acetate for 2–6 months (20 mg s.c., daily) before initiation of DMF therapy. Previous therapy was discontinued because of disease activity or side effects. DMF treatment was initiated 6 to 8 weeks later. The lymphocyte counts were in the normal range ($>1200$ μl$^{-1}$ PB) at the time-point of DMF therapy initiation. Blood was drawn as a baseline, 8 weeks after discontinuing from steroid treatment for treatment naïve patients, while for patients with previous therapies blood was drawn 6 to 8 weeks after discontinuing from IFN-β-1a or glatiramer acetate therapy. The second blood sample was drawn at the time point of disease progression defined by NEDA-3 criteria, mean for 18 patients: 10.8 months (Supplementary Tables 1–3). The study was conducted according to the rules of the Declaration of Helsinki. It was approved by the Ethics Committees of the University of Marburg and Mainz and all included patients gave written informed consent.

**Stimulation and staining of PBMCs from MS patient.** Peripheral blood mononuclear cells (PBMCs) were obtained from blood, which were subsequently frozen. Then, PBMCs were thawed, rested in pre-warmed RPMI/5% AB serum for 2 h, followed by restimulation with PMA (50 ng ml$^{-1}$)/Ionomycin (1 μg ml$^{-1}$), for 2.5 h and with addition of brefeldin A (5 μg ml$^{-1}$) for further 2.5 h. The cells were stained with: Zombie NIR fixable viability kit (#423106), anti-CD8a (SK1), anti-CD4 (RPA-T4), anti-CD14 (HCD14), anti-CD45RA (HI100), anti-IL-17A (BL168), all from Biolegend. Gated CD14$^-$CD4$^-$CD8$^+$CD45RA$^-$ or CD14$^-$CD8$^-$CD4$^+$ CD45RA$^-$ cells were analyzed for IL-17 on Aria III using Diva software (v8.0.2.). Gating strategy is included in Supplementary Fig. 1b. To control batch effects and day-to-day variations frozen PBMCs were used. For this, blood from a healthy donor was drawn, processed for PBMC isolation and then frozen in several aliquots which served as control samples. The control samples were thawed, rested, restimulated, fixed, stained, acquired and analyzed at the start of responder sample analysis (time point I) and then at two different days (time points II and III, for

non-responder analysis) using the same standard protocols, by the same person. The analysis reveals a very similar outcome, which gives mean for Tc17 frequency: $0.283 \pm 0.0153$ ($\pm$ sd) (Supplementary Fig. 8).

**Patient data analysis.** The analysis of IL-17 production by memory CD4$^+$ and CD8$^+$ T cells from PBMCs of MS patients before and after DMF therapy, who responded (NEDA-3 positive) or did not (NEDA-3 negative) to DMF was blinded and based on following power analysis. We performed power calculations, extrapolating from in vitro pilot experiments (IL-17 production by human memory CD8$^+$ T cells ± DMF treatment) to a patient estimate (Supplementary Fig. 1d).

We called relevant a difference of post-pre-DMF-treatment of 1.5 and detected a SD of 1.2 for the differences. With a significance level of alpha = 0.05 and an available patient pool of $n = 18$ (paired design, assuming normally distributed values), we would expect to obtain a power of about 95%.

We evaluated $P$ values for changes in the frequency of CD8$^+$CD45RA$^-$IL-17$^+$ cells through DMF therapy for each patient ($n = 1;…;18$, responder and non-responder), within the respective group, using two-tailed, paired $t$-tests ($P$ values shown above the groups, Fig. 1a). To compare the changes in the frequency of CD8$^+$CD45RA$^-$IL-17$^+$ cells through DMF therapy between responder and non-responder groups a two-tailed, unpaired Student's $t$-test was applied ($P$ value shown between groups, Fig. 1a). The change (Δ) in the frequency of CD8$^+$CD45RA$^-$IL-17$^+$ cells through DMF therapy, was calculated by subtracting the percentage of CD8$^+$ CD45RA$^-$IL-17$^+$ cells before DMF therapy ($X_1$) from the percentage of CD8$^+$ CD45RA$^-$IL-17$^+$ after therapy ($X_2$) for each patient ($\Delta = X_2 - X_1$). The same calculation was performed for the influence of DMF therapy on the frequency of CD4$^+$CD45RA$^-$IL-17A$^+$ T cells. $P$ values were determined using GraphPad version 8.0 software.

**Stimulation of CD8$^+$ T cells from healthy donors.** PBMCs from healthy donors were enriched by performing ficoll gradient centrifugation and then labeled with anti-CD8a (PRA-T8; BioLegend), anti-CD4 (OKT-4; BioLegend) and anti-CD45RA (HI100; BioLegend). CD8$^+$CD45RA$^-$ cells were sorted using a FACSAria™ III. Tc17 cells were stimulated with anti-CD3 (2 μg ml$^{-1}$, clone TR66) and anti-CD28 (2 μg ml$^{-1}$ CD28.2; BD Biosciences) in the presence of rh IL-1β 20 ng ml$^{-1}$, rh TGF-β 20 ng ml$^{-1}$, rh IL-6 10 ng ml$^{-1}$, rh IL-2 50 U ml$^{-1}$. Tc17 cells were treated with 0.1% DMSO as a control, or with 10 μM DMF alone or in combination with 50 μM GSH. Living cells were analyzed by flow cytometry for IL-17A Pacific Blue, anti-IL-17A (BL168; Biolegend) and IFN-γ APC-Cy7 anti-IFN-γ (4 S.B3; BioLegend) production after 4 days of culture and restimulation with PMA/ ionomycin for 2.5 h and brefeldin A for further 2.5 h. Supernatants were collected after 4 days and IL-17 was measured with the Human IL-17 DuoSet ELISA (R&D, DY317).

**Mice.** WT C57BL/6 were purchased from The Jackson Laboratory. Irf4$^{-/-}$, Tbx21$^{-/-}$, 2D2 mice expressing a transgenic TCR specific for MOG$_{35-55}$ and CD45.1$^+$ mice were bred at the Biomedical Research Center, University of Marburg. All mice were 8–12 weeks old, at C57BL/6 background and sex- and age-matched.

**Murine T cell purification and in vitro differentiation.** CD4$^+$ or CD8$^+$ T cell were obtained from LNs and spleens using negative selection kits (130–104–454 or 130–104–075) both from Miltenyi. For some experiments, CD4$^+$ or CD8$^+$T cells were sorted on Aria III (BD Biosciences) to obtain naïve CD44$^-$CD62L$^+$CD4$^+$ or CD44$^-$CD62L$^+$CD8$^+$T cells using anti-CD4V450 (RM4–5, BD Biosciences) or anti-CD8V500 (53–6.7, BD Biosciences), anti-CD44PE (IM7, eBiosciences) and anti-CD62LAF700 (MEL-14, BD Pharmingen) mAbs. Doublets were excluded through forward scatter-height by forward scatter-width and side scatter-high by side-scatter width parameters. Sorting purity was > 97% in post-sort analysis. Gating strategy is provided in Supplementary Fig. 9a. Tc17 were primed in RPMI (10%FCS) with plate-bound anti-CD3 mAb (5 μg ml$^{-1}$) and soluble anti-CD28 (0.5 μg ml$^{-1}$, both produced 'in-house'), rhIL-2 (50 U ml$^{-1}$, Novartis), rhTGF-β (0.5 ng ml$^{-1}$, Peprotech) rmIL-6 (30 ng ml$^{-1}$, Peprotech) and anti-IFN-γ (5 μg ml$^{-1}$, produced 'in-house'). For Th17 cell priming the conditions were used as for Tc17 cells with addition of rmIL-23 (20 ng ml$^{-1}$, Peprotech). Used inhibitors: DMF (Sigma-Aldrich, 242926, 20 μM), GSH (Sigma-Aldrich, G1404, 50 μM), Trolox (Merck Milipore, 648471, 400 μM), 2-DG (Millipore, 25972, 250 μM), rotenone (Sigma, R8875, 10 nM), oligomycin (Sigma, 04876, 15 nM) Akt1/2inhibitor (Akti, Merck Milipore, 124018, 1 μM), STAT5 inhibitor (STAT5i, Cayman, 15784, 35 μM), Ly294002 (Cell Signaling, 9901, 1 μM), TSA (Sigma-Aldrich, T8552, 1 nM) and rapamycin (Cell Signaling 9904, 50 nM).

**Flow cytometry.** For intracellular cytokine staining, murine cells were restimulated after 72 h of culture with PMA (50 ng ml$^{-1}$), Ionomycin (1 μg ml$^{-1}$, both from Sigma-Aldrich) and brefeldin A (5 μg ml$^{-1}$; Biolegend) for 4 h and fixed with 2% para-formaldehyde. Staining of transcription factors was performed without restimulation using the FOXP3/Fixation-Kit (eBioscience, 00–5521–00). Following antibodies were used for murine cell-analysis: anti-CD8a (eBiosciences, 53–6.7), anti-CD44 (eBiosciences, IM7), anti-CD62L (eBiosciences, MEL-14), anti-IFN-γ (Biolegend, XMG1.2), anti-IL-17A (eBiosciences, eBio17B7), anti-RORγt

(eBiosciences, B2D) or anti-T-BET (eBiosciences, eBio4B10). The cells were measured on FACSCalibur or FACSAria™ III (both BD) and analyzed with FlowJo Software (FlowJo LLC). For phospho-flow the cells were harvested after 48 h of culture, rested for 4 h and treated with 100U/ml rhIL-2 for 2 h, fixed and permeabilised using Lyse/Fix-Buffer (557870, BD) and Perm-BufferIII (558050, BD) then stained with anti-P-STAT5Y694 (eBiosciences, SRBCZX), anti-P-AKTS473 (Cell Signaling, D9E), anti-P-AKTY308 (Cell Signaling, D25E6), anti-P-FOXO1/3a (Cell Signaling, #9464) or anti-P-S6S235/236 (Cell Signaling, D57.2.2E).

**GSH Assay and ROS quantification**. Tc17 or Th17 cells were differentiated for 2 h ± DMF or DMF + GSH. Then, cells were washed and GSH/GSSG Glo™ Assay (Promega, V6611) was used or ROS levels were determined by incubation with 1 μM chloromethyl derivative of 2',7'-dichlorodihydrofluorescein diacetate (CM-H$_2$DCFDA, Thermo Fisher Scientific, C6827) according to the manufacturer's instructions.

**Extracellular Flux Assay**. For extracellular flux assay, CD8$^+$ cells were purified and differentiated to Tc17 cells for 3 days in the presence of DMSO or 20 μM DMF. On the day of assay 3–4 × 10$^5$ in vitro differentiated Tc17 cells were plated per well ($n = 6$) in a 96-well Seahorse plate in Assay media (Glucose-free DMEM supplemented with 2 mM glutamine, 1 mM NaCl, 0.5% phenol red, pH 7.35) in oxygen-free conditions. After 2 h of pre-incubation extracellular acidification rate (ECAR) was measured at baseline and in response to 10 mM glucose, 2.5 μM oligomycin and 100 mM 2-deoxy-D-glucose (2-DG) according to the manufacturer's protocols using the XF-96 Extracellular Flux Analyzer (Agilent).

**RNA-Seq and bioinformatics of murine Tc17 cells**. Tc17 cells obtained from naive CD62L$^+$CD44$^-$CD8$^+$ cells were cultured ± DMF or DMF + GSH for 48 h, then RNA was purified from Extrazol according to the manufacturer's specifications. RNA was purified with the RNeasy Plus Mini Kit according to the manufacturer's protocol (Qiagen). RNA was quantified with a Qubit 2.0 fluorometer (Invitrogen) and the quality was assessed on a Bioanalyzer 2100 (Agilent) using a RNA 6000 Nano chip (Agilent). Samples with an RNA integrity number (RIN) of >8 were used for library preparation. Barcoded mRNA-seq cDNA libraries were prepared from 400 ng of total RNA using NEBNext® Poly(A) mRNA Magnetic Isolation Module and NEBNext® Ultra™ RNA Library Prep Kit for Illumina® according to the manual. Quantity was assessed using Invitrogen's Qubit HS assay kit and library size was determined using Agilent's 2100 Bioanalyzer HS DNA assay. Barcoded RNA-Seq libraries were onboard clustered using HiSeq® Rapid SR Cluster Kit v2 using 8 pM and 50 bps were sequenced on the Illumina HiSeq2500 using HiSeq® Rapid SBS Kit v2 (50 Cycle). The raw output data of the HiSeq was preprocessed according to the Illumina standard protocol. Quality control on the sequencing data was performed with the FastQC tool (available at http://www.bioinformatics.babraham.ac.uk/projects/fastqc/), as well as the comprehensive Qorts suite. Inspecting the produced reports, all samples were deemed of good quality for further processing. Short reads alignment was performed with the ENSEMBL Mus_musculus.GRCm38 chosen as the reference genome. The corresponding annotation (ENSEMBL v76) was also retrieved from the ENSEMBL FTP website (http://www.ensembl.org/info/data/ftp/index.html). The STAR[66] aligner (version 2.4.0b) was used to perform mapping to the reference genome. Alignments were processed with the featureCounts[67] function of the Rsubread package, using the annotation file, also used for supporting the alignment. Exploratory data analysis was performed with the pcaExplorer package. Differential expression analysis was performed with DESeq2 package (version 1.18.1), setting the false discovery rate (FDR) to 0.01. The GLM framework of the DESeq2 R[68] package was applied, accounting for the sample preparation batch as a confounding factor in the model to increase the detection power. Gene expression profiles were plotted as heatmaps (color-coded z-scores for the regularized logarithm (rlog) transformed batch-corrected expression values.) with the R programming language and the pheatmap package (version 1.0.8). GSEA was performed as described[41]. Principal component analysis was performed using the pcaExplorer package (V2.5.1)[69].

**Patient cohort for transcriptome analysis**. To avoid individual-related differences we analyzed a matched cohort of 8 persons with MS diagnosed according to the McDonald criteria[63], which were stratified in two groups (Supplementary Fig. 2b): The first group termed "DMF untreated" consisted of 4 patients who were not treated with DMF before. In this group, three patients were treatment naive, meaning no treatment with any drug before, except of a short-term high dosage steroid treatment (1000 mg methylprednisolone for 5 consecutive days). Blood was drawn 8 weeks after discontinuing from steroid treatment. One patient was treated with Fingolimod (0.5 mg p.o daily) for four months and discontinued due to patient´s wish. Blood was drawn 6 months after Fingolimod discontinuation, at this time point the lymphocyte counts were in the normal range (>1200 μl$^{-1}$ PB). After blood collection, DMF therapy was initiated. The second group, termed "DMF treated" consisted of 4 patients who fulfilled NEDA-3 criteria after 12–13 months of DMF treatment (240 mg p.o twice a day). Before initiation of DMF therapy two patients were treatment naive, except of a short-term high dosage steroid treatment. DMF therapy was initiated 8 weeks after steroid discontinuation. Two patients were treated with IFN-β-1a (44 μg s.c., 3 times a week)

for 2–2.5 years before initiation of DMF therapy. One patient was discontinued from IFN-β 1a due to depression, whereas the second because of incompliance due to flu-like side effects. DMF therapy was initiated 4 to 6 months later. The lymphocyte counts were in the normal range (>1200 μl$^{-1}$ PB) at the time-point of DMF therapy initiation (patient characteristics are included in Supplementary Table 4). The PCA based on RNA-Seq data, revealed that the two group of patients (i) "DMF untreated" versus (ii) "DMF treated" clearly separate on the PC1 (Supplementary Fig. 2c). Since PC1 follows the treatment axis, the main driver of differences between these two groups was DMF treatment. This cohort was collected at the Department of Neurology at the University of Mainz. The study was conducted according to the rules of the Declaration of Helsinki. The Ethics Committee of the University of Mainz provided approval for this study and blood samples blood was drawn after written informed consent was obtained.

**RNA-Seq of human CD8$^+$ T cells from MS patients**. Fresh PBMCs from MS patients "DMF untreated" ($n = 4$) and "DMF treated" ($n = 4$) fulfilling NEDA-3 criteria after 12–13 months of DMF (Tecfidera) treatment (240 mg p.o twice a day) (Supplementary Table 4 and Supplementary Fig. 2b) were labeled with anti-CD8a-BV510 (SK1), anti-CD4-Pacifc Blue (RPA-T4), anti-CD14-FITC (HCD14), anti-CD45RA-BV650 (HI100), Zombie NIR; all from Biolegend. Memory CD8$^+$ T cells (CD8$^+$CD45RA$^-$) were sorted on FACSAria™III (BD Biosciences). Gating strategy is provided in Supplementary Fig. 9b. RNA was isolated using RNeasy plus micro kit (Qiagen), quantification, library preparation by using 20 ng of total RNA, sequencing (QC via FastQC and Qorts) was performed as described for mouse Tc17 cells. Sequencing reads were aligned with STAR (2.4.0j) to the ENSEMBL Homo_sapiens.GRCh38 reference genome, with annotation ENSEMBL v79. Similarly, the count matrix was obtained with featureCounts (Rsubread package). Differential expression analysis was performed with DESeq2 package (version 1.20), modeling the patient condition as only experimental factor (FDR = 0.01).

**Human IL-17$^+$CD8$^+$ and IL-17$^-$CD8$^+$ T cell gene signatures**. The RNA-Seq data (raw counts) from human IL-17$^+$CD8$^+$ and IL-17$^-$CD8$^+$ spleen cells from healthy donors was downloaded from the NCBI Gene Expression Omnibus under the accession number GSE96741. The contrast IL-17$^+$CD8$^+$ versus IL-17$^-$CD8$^+$ was calculated using the DESeq2 package[68], loaded from Bioconductor(version 3.9), with R (free software, version 1.1.423, Rstudio Inc v3.6). The differentially expressed genes ($p$ adj < 0.05), grouped into up- and down-regulation, were extracted as IL-17$^+$CD8$^+$ or IL-17$^-$CD8$^+$ gene signatures.

**Retroviral transduction**. Freshly isolated naïve CD8$^+$ T cells were plated in a 48-well plate (3 × 10$^5$ cells per well), 500 μl of retroviral supernatant with constitutive-active STAT5A1*6 (pMIG-STAT5)[70], or a control retrovirus (pMIG-empty) containing 7 μg ml$^{-1}$ polybrene and 50 U ml$^{-1}$ rhIL-2 was added and the cells were spun at 2700 rpm for 90 min at 37 °C. After spin infection, the cells were cultured under Tc17 conditions ± DMF for 2 h, then the cells were transfected for a second time as before, then cultured under Tc17 conditions ± DMF for 72 h, rested in RPMI + rhIL-2 and anti-mIFN-γ for further 72 h. Thereafter, the cells were re-cultured under Tc17 conditions for additional 72 h, then restimulated and analyzed by flow cytometry.

**Chromatin immunoprecipitation**. Tc17 cells were cultured for 72 h ± DMF or DMF + GSH as indicated in Figure legends. 2–5 × 106 cells were crosslinked with 1% formaldehyde for 6 min at room temperature subsequently, ChIP was performed. Lysed cells were sonicated in a Bioruptor® Plus (Diagenode) with 30s ON, 30s OFF on high power output for 27–33 cycles at 4 °C. For immunoprecipitation, 2.5–4 μg of the following Abs were used: anti-H4ac (Millipore, 06–866), anti-H3K4me3 (Active Motif, 39159), anti-H3K27me3 (Active Motif, 39155), anti-H3K27ac (Abcam, ab4729) or control IgG (Cell Signaling, 2729). Primer sequences for *Il17a promoter*, *Il17 enhancer-5*, *Il10 promoter* and *Rpl32* are provided in Supplementary Table 5. Amplifications were performed at the ABI Prism7500 (Applied Biosystems) using the Fast SYBR™Green (Thermo Fisher, 4385610). Values for non-specific binding (determined by control IgG) were subtracted. After normalization, the specific pulldown (input %) was calculated.

**Nuclear extraction and immunoblot**. Tc17 cells were differentiated for 72 h in the presence of DMSO ± 20 μM DMF ± 50 μM GSH. Cytosolic and nuclear lysates were produced using (10 mM HEPES pH7.9, 10 mM KCl and 1.5 mM MgCl$_2$) and (2% SDS and 66 mM Tris–HCl pH7.0), respectively. The following Abs were used: anti-H4Ac (Millipore, 06–866), anti-H3K4me3 (Active Motif, 39159), anti-H3K27me3 (Active Motif, 39155) and anti-H3K27Ac (Abcam, ab4729).

**EAE**. For induction of active EAE, C57BL/6 mice were immunized s.c. at the tail basis with 50 μg MOG$_{35-55}$ peptide (GenScript) emulsified in Complete Freund's adjuvant (CFA, BD) along with 100 ng pertussis toxin (PTX, List-Biological-Laboratories) administration i.p. on day 0 and 2. For therapeutic treatment, DMF was given daily (10 mg ml$^{-1}$) by oral gavage[26] in an emulsion 0.6% Methocel (methylcellulose, Sigma-Aldrich, M0262). Control animals were treated with 0.6% Methocel vehicle alone. Treatment started at day 8, after all animals exhibited

initial symptoms of EAE. For preventive treatment, DMF was applied by supplying the drinking water with 0.5 mg ml$^{-1}$ DMF 10 days before immunization. For adoptive transfer experiments $2.5 \times 10^6$ Tc17 ± DMF cells/mouse were injected i.p. ± 2D2 cells ($10^3$ cells per mouse) into $Irf4^{-/-}$ recipient mice one day before immunization. Daily clinical scoring of EAE symptoms was conducted as follows: 0, no symptoms, normal behavior; 1, tail paralyzed; 2, impaired righting reflex and gait; 3, partial hind limb paralysis; 4, hind legs completely paralyzed; 5, tetraparesis 6, dead. For the preparation of CNS lymphocytes, brains and spinal cords were excised and were dissociated for 40 min at 37 °C by digestion with collagenase D (0.5 mg/ml) and DNase I (10 µg/ml; both from Roche) in RPMI medium. Dispersed cells were passed through a 70-µm strainer and were pelleted by centrifugation, then were resuspended, layered onto a Percoll density gradient (GE Healthcare) and centrifuged for 30 min at 625 g and 22 °C. CNS lymphocytes were isolated by collection of the interphase fraction between 40% and 70% Percoll. After intensive washing in complete RPMI, cells were restimulated in vitro and were analyzed by flow cytometry. The cells were analyzed at the peak of disease (day 14–17 after immunization) for CD8a, CD4, CD45.1 (Biolegend, A20), IL-17A and IFN-y by flow cytometry.

**Statistical analysis**. Statistical analysis was performed using the GraphPad version 8.0 software. Data are presented as mean ± s.d. For all data sets, normality of distribution and homogeneity of variances was evaluated by Shapiro–Wilk test and Brown–Forsythe, respectively to test for violations of the assumptions inherent to parametric significance testing. None of the data sets showed significant departure from normality in the Shapiro–Wilk test. Statistical significance to compare two groups was evaluated using two-tailed/unpaired t-tests. In case of significant differences in variances between groups, Welch's correction was applied. The confidence interval was 95%. For multiple groups and/or multiple condition comparisons one-way or two-way analysis of variance (ANOVA) was performed followed by a Tukey's HSD, Dunnett's or Bonferroni post hoc test, respectively. In case of significant differences in variances between groups, Welch´s ANOVA with Games–Howell multiple comparison test was applied. A critical value for significance of $P < 0.05$ was used throughout the study, and statistical thresholds of 0.01, 0.001, as well as 0.0001 are indicated in the figures by asterisks (see legends for details). The exact P values, for ANOVA F values; for Welch's ANOVA, W values; t-test: t-values and degrees of freedom are provided in Supplementary Table 6 for main Figures and in Supplementary Table 7 for Supplementary Figures.

**Study approval**. All patients and controls gave their written informed consent after the Universities of Marburg and of Mainz IRB approval. Mouse experiments were approved by the local committee (Regierungspräsidium Gießen).

**Reporting summary**. Further information on research design is available in the Nature Research Reporting Summary linked to this article.

## Data availability

The data sets generated during and/or analyzed during the current study are available from the corresponding author on reasonable request. Sequencing data have been deposited in the Gene Expression Omnibus (GEO). The sequencing data is available under following accession numbers: the murine Affymetrix microarray data for CTL and Tc17 profiles: GSE110346, the murine RNA-seq data of Tc17 cells treated with DMSO, DMF or DMF + GSH: GSE116866, and the human data of MS patients before and after 12 months of DMF treatment: GSE116865.

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

## Acknowledgements

We thank Dr. Finkernagel, Dr. Green, Dr. Dolga, Dr. Jastroch, Dr. Bauer, Dr. Garn, M. Sc. Vogel, B. Sc. Erlemann, B. Sc. Kölbl, M. Sc. Ruhe, Mr. Happel, B. Sc. Girschik for experimental support. This work was supported by Biogen IIT-GER_BGT-13_10580 (M. H.;B.T.), UKGM 10/2016 (M.H.), DFG HU 1824/7-1 (M.H.), Fresenius Stiftung, 2015_A232 (M.H.), GIF I-1474-414.13/2018 M.H., M.B.), SFB/TR-128 (S.B.,F.Z.,F.C.K., T.B.), and SFB/TR-156 (F.C.K.); e:Med CAPSYS-FKZ01ZX1304E, JPIAMR Restrict-Pneumo-AMR - FKZ 01Kl1702 (B.S), SFB/TR-84 C1 (B.S), LOEWE Medical-RNomics-FKZ 519/03/00.001-(0003) (B.S); D.B. and L.B. are funded by FNR-PRIDE (PRIDE/11012546/NEXTIMMUNE), FNR-ATTRACT (A14/BM/7632103) and FNR-CORE (C15/BM/10355103); National Health and Medical Research Council (NHMRC) and Sylvia and Charles Viertel Foundation (A.K.); German Federal Ministry of Education and Research (BMBF 01EO1003). This study was investigator initiated and funded by an unrestricted grant of Biogen, the manufacturer of dimethyl fumarate (Tecfidera). Biogen was not involved in data storage, data analysis or data interpretation. Biogen was informed about the results and was neither involved in the submission, nor the publication of the paper.

## Author contributions

M.H., B.T., C.L., F.P., H.R., T.B., concept and design; C.L., H.R., L.C.C., A.G., Y.Z., R.G., L.B., M.G., F.M., F.P., F.S., W.B., M.K., S.M., F.C.K., Y.-Y.C., C.E.Z., B.S., V.S., A.K., M.L., D.B., M.B., F.Z., S.B., data acquisition, analysis and interpretation and M.H., B.T., C.L., F.P., H.R. manuscript preparation.

## Competing interests
The authors declare no competing interests.

## Additional information

Christina Lückel[1,2,23], Felix Picard [1,23], Hartmann Raifer[1,3,23], Lucia Campos Carrascosa[1,4], Anna Guralnik[1], Yajuan Zhang[1], Matthias Klein[2], Stefan Bittner [5], Falk Steffen[5], Sonja Moos[6], Federico Marini [7,8], Renee Gloury[9,10], Florian C. Kurschus[6,11], Ying-Yin Chao[12,13], Wilhelm Bertrams[14], Veronika Sexl[15], Bernd Schmeck[14,16], Lynn Bonetti[17], Melanie Grusdat[17], Michael Lohoff[1], Christina E. Zielinski[12,13], Frauke Zipp [5], Axel Kallies [9,10], Dirk Brenner[17,18,19], Michael Berger [20], Tobias Bopp [2,21], Björn Tackenberg[22] & Magdalena Huber[1]*

[1]Institute for Medical Microbiology and Hospital Hygiene, University of Marburg, 35043 Marburg, Germany. [2]Institute for Immunology, University Medical Center of the Johannes Gutenberg-University Mainz, 55131 Mainz, Germany. [3]Core-Facility Flow Cytometry, University of Marburg, 35043 Marburg, Germany. [4]Laboratory of Gastroentrology and Hepatology, Erasmus MC University Medical Center, 3015 CE Rotterdam, Netherlands. [5]Department of Neurology at the University Medical Center of the Johannes Gutenberg-University Mainz, 55131 Mainz, Germany. [6]Institute for Molecular Medicine, University Medical Center of the Johannes Gutenberg-University Mainz, 55131 Mainz, Germany. [7]Institute of Medical Biostatistics, Epidemiology and Informatics (IMBEI), University Medical Center of the Johannes Gutenberg-University Mainz, 55131 Mainz, Germany. [8]Center for Thrombosis and Hemostasis (CTH), University Medical Center of the Johannes Gutenberg-University Mainz, 55131 Mainz, Germany. [9]The Peter Doherty Institute for Infection and Immunity, Dept. of Microbiology and Immunology, University of Melbourne, Melbourne, VIC 3000, Australia. [10]The Walter and Eliza Hall Institute of Medical Research, 1 G Royal Parade, Parkville, VIC 3052, Australia. [11]Department of Dermatology, Heidelberg University Hospital, 69120 Heidelberg, Germany. [12]Center for Translational Cancer Research TranslaTUM, Technical University of Munich, 81675 Munich, Germany. [13]German Center for Infection Research (DZIF), Munich, Germany. [14]Institute for Lung Research, Universities of Giessen and Marburg Lung Center, Philipps-University Marburg, Member of the German Center for Lung Research (DZL), 35043 Marburg, Germany. [15]Institute of Pharmacology and Toxicology, University of Veterinary Medicine Vienna, 1210 Vienna, Austria. [16]Dept. of Medicine, Pulmonary and Critical Care Medicine, University Medical Center Giessen and Marburg, Philipps-University Marburg, Member of the German Center for Lung Research (DZL), 35043 Marburg, Germany. [17]Dept. of Infection and Immunity, Experimental and Molecular Immunology, Luxembourg Institute of Health, Esch-sur-Alzette L-4354, Luxembourg. [18]Luxembourg Centre for Systems Biomedicine (LCSB), University of Luxembourg, Belvaux, Luxembourg. [19]Odense Research Center for Anaphylaxis, Dept. of Dermatology and Allergy Center, Odense University Hospital, University of Southern Denmark, Odense DK-5000, Denmark. [20]The Lautenberg Center for Immunology and Cancer Research, IMRIC, Faculty of Medicine, The Hebrew University, Jerusalem 9112001, Israel. [21]Research Center for Immunotherapy (FZI), University Medical Center of the Johannes Gutenberg-University Mainz, 55131 Mainz, Germany. [22]Center of Neuroimmunology, Dept. of Neurology, University of Marburg, 35043 Marburg, Germany. [23]These authors contributed equally: Christina Lückel, Felix Picard, Hartmann Raifer. *email: magdalena.huber@staff.uni-marburg.de

