## [Peer Review File · Nature Communications]

Reviewers' comments:

Reviewer #2, clinical expert on MS (Remarks to the Author):

This article investigates the potential mechanism of DMF as immune modulatory treatment in MS based on the hypothesis that IL-17 producing cells are involved in the pathogenesis of MS.

I raise two main difficulties with this study in the way it is developed and presented;

1. The whole set of mechanistic elucidation work, per se quite extensive and thorough, emanates from the observation in fig. 1a. It is notable that although the change is statistically significant, the magnitude of the observed change is quite small. Mean or median values for the percentage of IL-17A producing cells are not provided in the figure but it looks like that from an average of less than 2% CD8+CD45RA- cells before treatment, the mean percentage is decreased to somewhere around or slightly below 1% at 12 month on therapy. The data is from 18 patients, but it looks like the significance in the reduction is largely driven from the small number of patients (4 or 5) who had >~2% IL-17A producing CD8+ memory cells. How significant could that be biologically? Furthermore, where is the evidence that this decrease is what actually mediates the therapeutic success in DMF-treated patients? While addressing biological significance is always difficult, ascertaining if clinical unresponsiveness to DMF treatment (in terms of evidence of disease activity, or non-NEDA status) is associated with failure to reduce IL-17A producing Tc17 cells is entirely feasible – especially since DMF treatment failure is fairly common, requiring treatment escalation.
2. The background evidence of involvement of Tc17 in MS is quite limited. Cited work includes an early paper by Tzartsos et al, but newer work linking IL-17 producing MAIT cells to MS pathology and pathophysiology in adult- as well as in paediatric MS is not considered in the manuscript. The authors could put in better context their studies and relate them to the full range of relevant, related literature.

Reviewer #3, expert on T cell responses in EAE (Remarks to the Author):

This manuscript characterizes the Dimethyl fumarate (DMF) treatment for multiple sclerosis and shows that this treatment reduced the frequency of IL-17-producing CD8+ T cells (Tc17). Based on the authors' previous work that Tc17 cells support Th17-mediated autoimmune encephalomyelitis (JCI), they concluded that DMF targets Tc17 to limit autoimmunity. Although the research is interesting, the conclusion is not solid and is not completely supported by the experimental data. There are the following concerns:

1. The percentage of Tc17 cells in the MS patient without DMF treatment was very low (not even flow data are shown) (Fig. 1a). The effect of DMF treatment in reducing the frequency of Tc 17 cells was thus very modest.
2. The authors compared the effect of DMF on murine Tc17 and Th17 cells, drawing the conclusion that DMF inhibited Tc17 cell differentiation, but not Th17 cells. According to the method they described, they used the same culture condition for Tc17 and Th17 cells (TGF- β +IL-6+IL-2). They did not use the pathogenic Th17 culture condition (TGF- β +IL-6+IL-23). Therefore, their result showing the effect of DMF on Th17 cells was not related to Th17 cells in the disease. In addition, there is a report in the literature showing that DMF prevents EAE by reducing TH1 and Th17 cell differentiation (PNAS, 2016, 113:4777), which also contradicts the current study. The authors did not discuss this difference.
3. The authors did test the effect of DMF on EAE. But it is not clear how they gave the DMF to EAE mice. Was it given starting with the day of immunization or at the onset of EAE? To mimic DMF

treatment in MS, DMF should be given to mice at the onset of EAE.

4. The authors stated that Tc17 cells could serve as a marker for the effectiveness of DMF treatment on MS; however, there were no data showing that DMF did not reduce the frequencies of Tc17 cells in MS patients who are refractory to such treatment.

Point-by-Point Responses to Reviewer Comments:

Reviewer #2 (Remarks to the Author):

1. The whole set of mechanistic elucidation work, per se quite extensive and thorough, emanates from the observation in fig. 1a. It is notable that although the change is statistically significant, the magnitude of the observed change is quite small. Mean or median values for the percentage of IL-17A producing cells are not provided in the figure but it looks like that from an average of less than 2% CD8⁺CD45RA⁻ cells before treatment, the mean percentage is decreased to somewhere around or slightly below 1% at 12 month on therapy. The data is from 18 patients, but it looks like the significance in the reduction is largely driven from the small number of patients (4 or 5) who had >~2% IL-17A producing CD8⁺ memory cells. How significant could that be biologically? Furthermore, where is the evidence that this decrease is what actually mediates the therapeutic success in DMF-treated patients? While addressing biological significance is always difficult, ascertaining if clinical unresponsiveness to DMF treatment (in terms of evidence of disease activity, or non-NEDA status) is associated with failure to reduce IL-17A producing Tc17 cells is entirely feasible – especially since DMF treatment failure is fairly common, requiring treatment escalation

Thank you for the valuable comment and appreciation of our mechanistic work. In Fig. 1a we provided graphs with connected lines (before-after) visualising analysis of the same patient. We apologize for not including mean values, we plotted the data as suggested, which shows that the mean for CD8⁺CD45RA⁻IL-17⁺ cells before DMF therapy from 18 responders (NEDA-3 positive patients) is indeed lower than 2%, it is 1.23 ± 0.99 (mean \pm SD, n=18), after therapy it falls to 0.51 ± 0.42 , resulting in a difference between means=0.72, P= 0.001, two-tailed paired t-test (point-by-point reply, Fig. 1a). Before therapy initiation, higher frequencies of Tc17 cells were detectable in 8 patients, whereas lower frequencies in 10 patients, indicating a variability within the responder cohort (Supplementary Table 2). To understand the biological significance of the Tc17 reduction by DMF therapy in responders, we analysed non-responders (NEDA-3-negative patients, n=18), as suggested. In this cohort the frequency of Tc17 cells was not significantly changed upon DMF therapy, the mean before is 0.27 ± 0.19 , whereas 0.30 ± 0.35 after, difference between means=0.033, P=0.72, two-tailed paired t-test (point-by-point reply, Fig. 1c). The comparison of reduction in percentages of CD8⁺CD45RA⁻IL-17⁺ (after vs before DMF therapy) between responders and non-responders was significant P=0.002 (two tailed unpaired t-test, n=18), indicating that the suppression of Tc17 cells accompanies a positive response to DMF (manuscript, modified Fig. 1a). Into the new version of the manuscript we included gating strategy as a new Suppl. Fig. 1b, to visualize how we obtained the data presented in the modified Fig. 1a as well as new Fig. 1b and Supplementary Fig. 1c.

Figure 1. Suppression of IL-17 production by memory CD8⁺ T cells accompanies positive response to DMF in MS. Mean values for Tc17 and Th17 frequency in responders and non-responders. Flow cytometry of IL-17A in CD14⁻CD45RA⁻CD8⁺ or CD14⁻CD45RA⁻CD4⁺ cells from peripheral blood of the same MS patients before and after DMF therapy fulfilling (Responders,

a,b, n=18) or not (Non-Responders, c,d, n=18) NEDA-3 criteria The observer was blinded to experimental groups. Data are shown as mean \pm SD, n=18. p-values evaluated by two-tailed, paired t-test,

Interestingly, the responder versus non-responder cohort significantly differed in the abundance of Tc17 cells before the therapy (manuscript new Fig. 1b), suggesting that even low frequencies which are detectable in peripheral blood (PB) can potentially relate to disease mechanisms. We presume that the DMF responder group consists of MS patients, in which Tc17 cells might be involved in disease pathogenesis. Furthermore, our data suggests that increased frequency of Tc17 cells could be one of the parameters which could predict positive response to DMF. However, having low frequencies of Tc17 cells doesn't predict negative response, since within responders there is a subgroup of ten patients with low Tc17 abundance (manuscript Fig. 1b). Further studies are needed to evaluate the response mechanisms to DMF in this subgroup of responders.

Although meeting key eligibility criteria as described in Method section, the non-responder group is more heterogeneous as compared to responder group in terms of previous treatment and EDSS (manuscript, Supplementary Tables 1-3). In the non-responder group, we included 10 treatment naïve patients with EDSS 0.9 and 8 patients with EDSS 2.8 (prior therapy 4-12 months with IFN- β 1a or glatiramer acetat) at the therapy start. We provide tables with clinical characteristics and percentages of CD8⁺CD45RA⁻IL-17⁺T cells before and after therapy for each responder and non-responder (manuscript, Supplementary Tables 1-3). To understand, if EDSS score influenced the frequency of Tc17 cells before and upon DMF treatment, we compared the frequencies of CD8⁺CD45RA⁻IL-17⁺ cells before and after therapy initiation as well as their reduction between these two subgroups of patients (EDSS 0.9 vs EDSS2.8, point-by-point-reply Fig. 2a). Importantly, there was no significant difference, indicating that both subgroups responded similarly to DMF treatment in terms of frequency of CD8⁺CD45RA⁻IL-17⁺ cells. Furthermore, the abundance of CD8⁺CD45RA⁻IL-17⁺T cells before and after DMF therapy did not significantly differ between these two subgroups (point-by-point-reply Fig. 2b,c), indicating that EDSS score of non-responders didn't significantly influence the response to DMF in terms of CD8⁺CD45RA⁻IL-17⁺ cells. Therefore, we believe that the comparative data between the responder and non-responder group for IL-17⁺CD8⁺CD45RA⁻ T cells is valid.

Fig. 2. EDSS score does not significantly influence the frequency and response of Tc17 cells to DMF within the non-responder cohort. Flow cytometry of IL-17A in CD45RA⁻CD8⁺ cells from PB of the same MS patients before and after DMF therapy not fulfilling NEDA-3 criteria after DMF therapy (Non-Responders). The cohort was subdivided

into two groups characterized by specific EDSS, the treatment naïve group (n=10) had EDSS of 0.9 at the therapy start, whereas the group with previous treatment had EDSS 2.8 at the therapy start. **a**, Compared are the frequencies of Tc17 cells from the same patient before and after DMF therapy, *P* values were calculated by two-tailed, paired t-test. Comparison in the reduction in Tc17 frequencies between two subgroups of DMF non-responders (EDSS 0.9 versus EDSS 2.8), *P* value was calculated by two-tailed, unpaired t-test (*p*=0.59). **b**, **c** Comparison in frequencies of CD8⁺CD45RA⁻IL-17⁺cells before (**b**) and after (**c**) DMF therapy between two subgroups of DMF non-responders (EDSS 0.9 versus EDSS 2.8), *p*-values were calculated by two-tailed, unpaired t-test. The observer was blinded to experimental groups. **b**, **c**, Data are shown as mean±SD.

Further support for biological significance of our results provided comparison of recently published gene expression profiles from sorted CD3⁺CD4⁻CD8⁺Vα7.2TCRγδ⁻IL-17⁺ and CD3⁺CD4⁻CD8⁺Vα7.2TCRγδ⁻IL-17⁻ from spleens of healthy individuals¹ with our data of CD45RA⁻CD8⁺ T cells from DMF-untreated versus treated MS patients. Applying GSEA, we found that in cells from DMF-untreated patients CD8⁺IL-17⁺ associated genes were enriched, conversely, in cells from DMF-treated patients CD8⁺IL-17⁻ associated genes were more abundant (manuscript, new Fig. 2k,l, new Supplementary Fig. 2f,g), revealing that the expression of human IL-17⁺CD8⁺ signature genes distinguished DMF-treated versus untreated patients. This data indicates that DMF therapy suppresses not only IL-17 but also Tc17-associated genetic profile in therapy responders. Moreover, the mechanistic experiments *in vitro* (Fig. 3-5) and *in vivo* in the mouse model in therapeutic setting (new Fig. 6a-d) as well as adoptive transfers (Fig. 6e-k) corroborate the associative human data.

Finally, IL-17-producing CD4⁺ T cells (Th17) cells, which are commonly accepted as contributors to MS, were detectable in the responder group in the frequency of 1.02±1.63 (mean±SD, n=18) before treatment, while 0.75±0.92 after, difference between means=0.27, P value: 0.34, two-tailed paired t-test (point-by-point reply, Fig. 1c). In non-responders the frequency of Th17 cells were as follows: 0.32±0.37 (mean±SD, n=18) before treatment, while 0.21±0.21 after, difference between means -0.09, P value: 0.35, two-tailed paired t-test (point-by-point reply, Fig. 1d). Interestingly, Th17 cell frequencies before DMF treatment in the responder versus non-responder group did not significantly differ (manuscript, new Supplementary Fig. 1c), further supporting the idea on the involvement of Tc17 cells in the DMF-mediated immune modulation. In terms of percentages, similar percentages of Th17 cells in PB of MS patients were detected in a recently published Nature Communications study by H. L. Weiner². For another population of IL-10⁺IFNγ⁻ CD4⁺ T cells in MS patients published in Nature Immunology last year by D. Hafler, the percentages in PB were also at approx. 1%³. This indicates, that the percentages we detected in the periphery were also detectable by other groups and although low they can have acknowledged biological significance.

Overall, we believe our data provides a relevant and new insight into the mechanisms of the CNS autoimmunity on immunomodulatory action of DMF. We thank the reviewer for the advice to analyse Tc17 and Th17 cell frequency in DMF non-responder group, since this data strengthen our conclusions.

2. The background evidence of involvement of Tc17 in MS is quite limited. Cited work includes an early paper by Tzartsos et al, but newer work linking IL-17 producing MAIT cells to MS pathology and pathophysiology in adult- as well as in paediatric MS is not considered in the manuscript. The authors could put in better context their studies and relate them to the full range of relevant, related literature.

We thank the reviewer for the stimulating comment and apologize for omission of MAIT and additional Tc17 literature. In the new version of the manuscript we included newer literature on IL-17 producing cells including classical Tc17 and MAIT cells and following paragraphs in the introduction and discussion section:

Introduction, raw 73-87, page 4:

“Notably, CD8⁺ T cells are found in higher frequency than CD4⁺ T cells⁴⁻⁶ and primarily among CD8⁺ T cells large clonal expansions have been reported in active demyelinating MS lesions⁷. IL-17-producing CD8⁺ (Tc17) are enriched in cerebrospinal fluid (CSF) in early MS⁸ and Tc17 frequencies in CSF correlate with disability⁹. Furthermore, increased frequencies of Tc17 cells were detected in peripheral blood (PB) of MS patients as compared to healthy controls¹⁰ and Tc17 cells were present in active areas of acute and in chronic MS lesions alongside with IL-17-producing CD4⁺ (Th17) T cells¹¹,

implicating a contribution of both subpopulations to MS pathogenesis. Interestingly, many of IL-17-producing CD8⁺ T cells in MS patients bear features of mucosal associated invariant T (MAIT) cells, which are MHC-related protein 1 (MR1)-restricted CD8⁺ T cells dependent on commensal microbiota¹²⁻¹⁶. Functionally, using experimental autoimmune encephalomyelitis (EAE) as a pre-clinical mouse model for MS, we showed that Tc17 cells provided “reverse help” for the encephalogenicity of IL-17-producing CD4⁺ T (Th17) cells via their hallmark cytokine IL-17A⁸, revealing an important Tc17-dependent enhancement of Th17-mediated autoimmunity of the CNS.”

Discussion, raw 345-359, page 17 and 18:

“In MS patients the majority of IL-17-producing CD8⁺ T cells expresses the molecules CD161 and CCR6 as well as TCRV α 7.2, characterizing them as MAIT cells^{13, 14, 16}. Functionally, these cells seem to have pathogenic relevance since paediatric MS patients harboured more IL-17 producing MAIT cells in PB, as compared to healthy controls or children with monophasic inflammatory CNS disorder¹⁵. Furthermore, increased IL-17 production by MAIT cells¹⁷ as well as their enhanced accumulation in brain lesions¹⁶ was detectable in MS. Considering these reports, it is possible, that the DMF-mediated reduction of IL-17 producing CD8⁺ T cells in MS patients, which we herein described, may also affect CD161^{high} CD8⁺ MAIT cells. As MAIT cells were not affected by IFN- β therapy, but strongly reduced by high dose of cyclophosphamide in combination with alemtuzumab treatment followed by autologous stem cell transplantation in MS patients¹⁴, one can speculate that besides the non-myeloablative depletion, MAIT cells might be susceptible to ROS upregulated by DMF-mediated glutathione depletion. Therefore, it will be of interest for future studies to compare in details the susceptibility to DMF of conventional versus CD161^{high} CD8⁺ MAIT Tc17 cells in patients, to define the main target population within IL-17-producing CD8⁺ T cells and to closely characterize the therapy responder group.”

This reviewer comment stimulated us to analyse MAIT cells in MS patients upon DMF therapy. We established MAIT staining (point-by-point-reply Fig. 3a) and analysed Tc17 and MAIT cell frequencies in three available frozen PBMC samples from DMF responders (before and after one year of therapy). The results show that also IL-17-producing MAIT cells were suppressed by DMF treatment at least in the analysed three available samples (point-by-point-reply Fig. 3b). To understand the contribution of MAIT cells to the described effect of DMF on Tc17 cells in MS patients, we applied (i) exclusion gating in this three samples and (ii) analysis of gene expression profiles of Tc17 cells from DMF-treated and untreated patients using published data set for human IL-17-producing MAIT cells¹⁸. Exclusion gating revealed that MAIT cells contributed to the DMF effect on Tc17 cells (point-by-point-reply Fig. 3a). GSEA of gene expression profiles revealed a tendency towards enrichment of MAIT profile in CD8⁺CD45RA⁻ cells from DMF-untreated MS patients (point-by-point-reply Fig. 3c, d), further supporting the idea that IL-17-producing MAIT cells contributed to the described DMF effect on Tc17 cells to some extent. However, this issue should to be analysed in detail in future, we included this speculation into the discussion section.

Fig. 3. Frequency of MAIT and Tc17 cells in DMF responders before and after therapy. a, gating strategy for MAIT cells. Acquired cells were first gated for exclusion of debris (FSC-H vs SSC-H), viable cells were identified using LDS-NIR and gating LDS-NIR⁻ cells, then for singlets (FSC-A vs FSC-H), then monocytes were excluded by gating on CD14⁻ cells. Memory CD8⁺ T cell subset was identified by gating of CD45RA⁻CD8⁺ cells, by which IL-17 and IFN- γ positivity was analyzed. MAIT cells were identified by gating on CD161⁺TCR α 7.2⁺ cells. **b**, Flow cytometry of IL-17A in CD14⁻CD45RA⁻CD8⁺ or MAIT cells from peripheral blood of the same MS patients (n=3, patient characteristics point-by-point reply Table 1) before and after DMF therapy fulfilling NEDA-3. Individual values are plotted and mean is shown. **c**, GSEA of genes associated with MAIT profile, with 165 genes upregulated in MAIT cells compared to TCR α 7.2⁻ conventional T cells, provided by Park et al¹⁸, in CD8⁺CD45RA⁻

T cells from untreated vs treated MS patients. d, Heatmap of color coded z-scores for the rlog transformed expression values based on the GSEA comparing the relative expression of genes in CD8⁺CD45RA⁻ T cells from “DMF-untreated” versus matched “DMF-treated” MS patients examining the distribution of MAIT signature genes according to Park et al¹⁸

Supplementary Table 1. Characteristics of MS patients included for IL-17A analysis by MAIT and CD8+CD45RA-cells (point-by-point reply Fig. 3).

Clinical data	Responders (n=3)
Age at MS ¹ onset, years: mean; SD; range)	27; 6.9; 23-35
F:M (n/n; ratio)	2/1; 2
Duration of MS since diagnosis before DMF therapy initiation (months: mean; SD; range)	2; 0; 2
Previous therapies	None (3)
DMF treatment (months: mean; SD)	12.7; 1.15
EDSS (mean; SD; range) before DMF therapy	0.33; 0.58; 0–1
EDSS (mean; SD; range) after DMF therapy	0.33; 0.58; 0–1

¹: according to McDonald criteria (Polman et al., 2011)

Reviewer #3 (Remarks to the Author):

This manuscript characterizes the Dimethyl fumarate (DMF) treatment for multiple sclerosis and shows that this treatment reduced the frequency of IL-17-producing CD8+ T cells (Tc17). Based on the authors' previous work that Tc17 cells support Th17-mediated autoimmune encephalomyelitis (JCI), they concluded that DMF targets Tc17 to limit autoimmunity. Although the research is interesting, the conclusion is not solid and is not completely supported by the experimental data. There are the following concerns:

1. The percentage of Tc17 cells in the MS patient without DMF treatment was very low (not even flow data are shown) (Fig. 1a). The effect of DMF treatment in reducing the frequency of Tc 17 cells was thus very modest.

4. The authors stated that Tc17 cells could serve as a marker for the effectiveness of DMF treatment on MS; however, there were no data showing that DMF did not reduce the frequencies of Tc17 cells in MS patients who are refractory to such treatment.

Thank you for these valuable comments. Please see our reply to Reviewer #2, point 1.

2. The authors compared the effect of DMF on murine Tc17 and Th17 cells, drawing the conclusion that DMF inhibited Tc17 cell differentiation, but not Th17 cells. According to the method they described, they used the same culture condition for Tc17 and Th17 cells (TGF- β +IL-6+IL-2). They did not use the pathogenic Th17 culture condition (TGF- β +IL-6+IL-23). Therefore, their result showing the effect of DMF on Th17 cells was not related to Th17 cells in the disease.

Thank you for the comment. Indeed, in the previous version of manuscript we included data from Th17 cells treated with TGF- β +IL-6+IL-2. We removed this data from the new version and replaced them with new data from Th17 cells cultured under TGF- β +IL-6+IL-23+IL-2 conditions (new Fig. 1f, g), which confirm the results obtained with non-pathogenic Th17 cells. Moreover, we replaced the data with inhibitors in the Supplementary Fig. 3b and 3f with results from Th17 cells differentiated under TGF- β +IL-6+IL-23+IL-2 conditions (modified Supplementary Fig. 3b and 3f). Interestingly, AKT inhibitor in the presence of DMF inhibited IL-17 production in Th17 cells differentiated under TGF- β +IL-6+IL-23+IL-2 conditions (Supplementary Fig. 3b), this is in strong contrast to Tc17 cells, in which AKT inhibitor boosted IL-17 production (Fig. 3g). This result supports the idea on differential regulation of IL-17 production by Th17 and Tc17 cells, which is in agreement with the recent publication showing specific transcriptional profiles for mouse and human Tc17 and Th17 cells¹. We refer to this publication in the context of Th17/Tc17 cells in the discussion section in the following way:

Discussion, raw 397-400, page 19-20:

“Hence, our data indicate cell-type specific signaling pathways controlling IL-17 production, which could explain particular responsiveness of Tc17 to DMF treatment. This is in agreement with a recent publication demonstrating specific transcriptional programs for mouse and human Tc17 and Th17 cells¹.”

To further understand the influence of DMF on subpopulations of Th17 cells, we include analysis Th17 cells differentiated under IL-1 β +IL-6+IL-23+IL-2 conditions, which has been also described as pathogenic Th17 cells¹⁹. In this Th17 cell subpopulation there was a tendency towards inhibition of

IL-17 by DMF, however non-significant, although significant ROS upregulation (point-by-point reply, Fig. 4a,b), confirming the data obtained on non-pathogenic (old version of the manuscript) and pathogenic Th17 cells (new Fig 1f, g, Supplementary Fig. 3b, f). There was again a difference in the action of AKT inhibitor in connection with DMF, as for non-pathogenic Th17 cells, this drug combination did not significantly impact IL-17 production by Th17 cells differentiated in the presence of IL-1 β -IL-6+IL-23+IL-2. Overall these results argue for differential impact of DMF on Th17 and Tc17 cells, and on specific regulation of IL-17 production in different subtypes of T cells.

Figure 4. DMF upregulates ROS in Th17 while the influence on IL-17 production is independent on AKT. **a**, Flow cytometry of ROS levels in pathogenic Th17 cells (conditions: IL-6+IL-1 β +IL-23+IL-2) differentiated for 2h as determined by CM-H₂DCFDA staining (fold of geometric mean fluorescence intensity (MFI), normalized to the corresponding control, which was arbitrarily set to 1. **b-d**, Flow cytometry of IL-17A in Th17 cells differentiated as in (a) for 72h with indicated treatment. Bars show mean \pm s.d. from four combined experiments. Bars show mean \pm s.d. from four (a-d) combined experiments, * p <0.05 for (a) by two-tailed, unpaired t -test, for (b) by one-way ANOVA followed by Tukey's Honestly Significant Difference (HSD) multiple comparison test.

In addition, there is a report in the literature showing that DMF prevents EAE by reducing TH1 and Th17 cell differentiation (PNAS, 2016, 113:4777), which also contradicts the current study. The authors did not discuss this difference.

Thank you for the comment. In the old version of our manuscript we already cited the publication by Schulze-Topphoff et al, we apologize for omitting adequate discussion. Our data in adoptive transfer model show that DMF-treated Tc17 cells, cause mild EAE (manuscript Fig 6e), reflected in lower T cell number in CNS (manuscript Fig. 6f) and decreased frequencies of Th17 cells in CNS (manuscript Fig. 6k). These results are in agreement with the data published by Schulze-Topphoff on reduction of Th17 cell frequency in CNS by DMF *in vivo*. We believe that modulation of Th1 cells, monocytes, macrophages and dendritic cells also belongs to the immune modulatory activity of DMF, besides its influence on Tc17 and *in vivo* on Th17 cells. We included a following paragraph on this topic into the discussion section:

Discussion, raw 329-344, page 16-17:

“In the mouse model, oral DMF treatment in therapeutic as well as in preventive setting ameliorated clinical signs of disease and suppressed frequency of IL-17-producing CD8⁺ T cells in CNS, consistent with the results obtained from PB of MS patients. In adoptive transfer model, the amelioration of the disease by DMF was caused by a stable suppression of Tc17 cells and thereby loss of their co-pathogenic function resulting in the reduced frequency of Th17 cells in the CNS. This is consistent with previous reports showing reduced frequencies of Th17 cells in DMF treated mice^{20, 21}. Immune modulatory effects of DMF also include influence on IFN- γ production by CD4⁺ T cells^{20, 21} as well as on the phenotype of dendritic cells, monocytes^{20, 22, 23} and metabolism of macrophages²¹, which likewise contribute to the its therapeutic effect. Considering multiple mechanisms driven by DMF

and an extensive heterogeneity in the disease course resulting from distinct effector mechanisms underlying MS²⁴, we believe that our findings and conclusions apply to a subset of patients, in which Tc17 cells are involved in the disease pathogenesis. This idea is supported by our finding that the mean frequency of Tc17 cells before DMF therapy was significantly higher in responders as compared to non-responders however, further studies should prove this concept.”

The authors did test the effect of DMF on EAE. But it is not clear how they gave the DMF to EAE mice. Was it given starting with the day of immunization or at the onset of EAE? To mimic DMF treatment in MS, DMF should be given to mice at the onset of EAE.

Thank you for bringing up this important point. In the old version of our manuscript we applied the protocol published by Ghoreschi et al²², in which DMF was given in drinking water already 10 days before immunization till the end of the experiment (new Supplementary Fig. 6b-d). As suggested, we performed a new experiment with therapeutic DMF oral application (daily oral gavage) starting from disease onset (day 8) using dosage of 100mg/kg as previously described by Schulze-Topphoff et al²⁰. We included the data into the new version of the manuscript as a new Fig 6a-d and new Supplementary Fig. 6a. The results show disease amelioration by DMF treatment, reflected in significantly decreased T cell infiltration in CNS as well as in decreased frequency of Tc17 cells as compared to untreated animals. This data further corroborates the idea on the suppression of Tc17 by DMF *in vivo*. We thank the reviewer for raising this important point which helped us to strengthen conclusions of our manuscript.

References:

1. Mielke LA, *et al*. TCF-1 limits the formation of Tc17 cells via repression of the MAF-RORgammat axis. *J Exp Med*, (2019).
2. Hu D, *et al*. Transcriptional signature of human pro-inflammatory TH17 cells identifies reduced IL10 gene expression in multiple sclerosis. *Nat Commun* **8**, 1600 (2017).
3. Sumida T, *et al*. Activated beta-catenin in Foxp3(+) regulatory T cells links inflammatory environments to autoimmunity. *Nat Immunol* **19**, 1391-1402 (2018).
4. Booss J, Esiri MM, Tourtellotte WW, Mason DY. Immunohistological analysis of T lymphocyte subsets in the central nervous system in chronic progressive multiple sclerosis. *J Neurol Sci* **62**, 219-232 (1983).
5. Hauser SL, Bhan AK, Gilles F, Kemp M, Kerr C, Weiner HL. Immunohistochemical analysis of the cellular infiltrate in multiple sclerosis lesions. *Ann Neurol* **19**, 578-587 (1986).
6. Machado-Santos J, *et al*. The compartmentalized inflammatory response in the multiple sclerosis brain is composed of tissue-resident CD8+ T lymphocytes and B cells. *Brain* **141**, 2066-2082 (2018).
7. Babbe H, *et al*. Clonal expansions of CD8(+) T cells dominate the T cell infiltrate in active multiple sclerosis lesions as shown by micromanipulation and single cell polymerase chain reaction. *J Exp Med* **192**, 393-404 (2000).

8. Huber M, *et al.* IL-17A secretion by CD8+ T cells supports Th17-mediated autoimmune encephalomyelitis. *J Clin Invest* **123**, 247-260 (2013).
9. Lolli F, *et al.* Increased CD8+ T cell responses to apoptotic T cell-associated antigens in multiple sclerosis. *J Neuroinflammation* **10**, 94 (2013).
10. Wang HH, *et al.* Interleukin-17-secreting T cells in neuromyelitis optica and multiple sclerosis during relapse. *J Clin Neurosci* **18**, 1313-1317 (2011).
11. Tzartos JS, *et al.* Interleukin-17 production in central nervous system-infiltrating T cells and glial cells is associated with active disease in multiple sclerosis. *Am J Pathol* **172**, 146-155 (2008).
12. Korn T, Kallies A. T cell responses in the central nervous system. *Nat Rev Immunol* **17**, 179-194 (2017).
13. Annibaldi V, *et al.* CD161(high)CD8+T cells bear pathogenetic potential in multiple sclerosis. *Brain* **134**, 542-554 (2011).
14. Abrahamsson SV, *et al.* Non-myeloablative autologous haematopoietic stem cell transplantation expands regulatory cells and depletes IL-17 producing mucosal-associated invariant T cells in multiple sclerosis. *Brain* **136**, 2888-2903 (2013).
15. Mexhitaj I, *et al.* Abnormal effector and regulatory T cell subsets in paediatric-onset multiple sclerosis. *Brain* **142**, 617-632 (2019).
16. Willing A, *et al.* CD8(+) MAIT cells infiltrate into the CNS and alterations in their blood frequencies correlate with IL-18 serum levels in multiple sclerosis. *Eur J Immunol* **44**, 3119-3128 (2014).
17. Willing A, Jager J, Reinhardt S, Kursawe N, Friese MA. Production of IL-17 by MAIT Cells Is Increased in Multiple Sclerosis and Is Associated with IL-7 Receptor Expression. *J Immunol* **200**, 974-982 (2018).
18. Park D, *et al.* Differences in the molecular signatures of mucosal-associated invariant T cells and conventional T cells. *Sci Rep* **9**, 7094 (2019).
19. Lee Y, *et al.* Induction and molecular signature of pathogenic TH17 cells. *Nat Immunol* **13**, 991-999 (2012).
20. Schulze-Topphoff U, *et al.* Dimethyl fumarate treatment induces adaptive and innate immune modulation independent of Nrf2. *Proc Natl Acad Sci U S A* **113**, 4777-4782 (2016).

21. Kornberg MD, *et al.* Dimethyl fumarate targets GAPDH and aerobic glycolysis to modulate immunity. *Science*, (2018).
22. Ghoreschi K, *et al.* Fumarates improve psoriasis and multiple sclerosis by inducing type II dendritic cells. *J Exp Med* **208**, 2291-2303 (2011).
23. Carlstrom KE, *et al.* Therapeutic efficacy of dimethyl fumarate in relapsing-remitting multiple sclerosis associates with ROS pathway in monocytes. *Nat Commun* **10**, 3081 (2019).
24. Dendrou CA, Fugger L, Friese MA. Immunopathology of multiple sclerosis. *Nat Rev Immunol* **15**, 545-558 (2015).

List of new added main figures

Figure	Figure shows	Page inserted
1 b	Frequency of CD8 ⁺ CD45RA ⁻ IL-17A ⁺ cells before DMF therapy in Responders and Non-Responders.	7
1 f	Flow cytometry of ROS levels in pathogenic Th17 cells.	8
1 g	Flow cytometry of IL-17A in pathogenic Th17 cells.	8
2 k	GSEA of genes associated with IL17 ⁺ CD8 ⁺ profiles in CD8 ⁺ CD45RA ⁻ T cells from DMF treated vs untreated MS patients.	10-11
2 l	GSEA of genes associated with IL17 ⁺ CD8 ⁺ profiles in CD8 ⁺ CD45RA ⁻ T cells from DMF treated vs untreated MS patients.	10-11
6 a	Mean clinical scores of WT mice \pm therapeutic DMF application.	14
6 b	T cell numbers in the CNS of WT mice \pm therapeutic DMF application.	14
6 c	CD8 ⁺ T cell numbers in the CNS of WT mice \pm therapeutic DMF application.	14
6 d	Percentages of IL-17A ⁺ CD8 ⁺ T cells in the CNS of WT mice \pm therapeutic DMF application.	14

List of incorporated changes in main figures

Old figure label	New figure label	Changes to figure	Page mentioned
1 a	1 a	Flow cytometry from DMF Non-Responders	7
1 b	1 c		7
1 c	1 d		8
1 d	1 e		8
1 e		Removed from manuscript	
1 f		Removed from manuscript	
1 g	1 h		8
1 h	1 i		9
1 i	1 j		9
6 a	Suppl. 6 c		14
6 b	Suppl. 6 d		14
6 c	6 e		14
6 d	6 f		14
6 f	6 h		15
6 g	6 i		15
6 h	6 j		15
6 i	6 k		15

List of new added Supplementary figures

Figure	Figure shows	Page inserted
1 b	General flow cytometry gating strategy for analysis of the frequencies of IL-17 ⁺ cells among CD45RA ⁺ CD8 ⁺ as well as CD45RA ⁺ CD4 ⁺ T cells	7
1 c	Frequency of CD4 ⁺ CD45RA ⁺ IL-17A ⁺ cells before DMF therapy in Responders versus Non-Responders	7
2 f	Heatmap of relative expression of IL17 ⁺ CD8 ⁺ genes in CD8 ⁺ CD45RA ⁺ T cells from DMF untreated vs treated MS patients	10-11
2 g	Heatmap of relative expression of IL17 ⁺ CD8 ⁺ genes in CD8 ⁺ CD45RA ⁺ T cells from DMF untreated vs treated MS patients	10-11
3 b	Flow cytometry of IL-17A in pathogenic Th17 cells	12
3 f	Flow cytometry of IL-17A in pathogenic Th17 cells	12
6 a	Experimental design for EAE with therapeutic DMF application	14
6 b	Experimental design for EAE with preventive DMF application	14
6 e	Outline of experimental strategy for adoptive transfer EAE	14

List of incorporated changes in Supplementary figures

Old figure label	New figure label	Changes to figure	Page mentioned
1 b	1 d		8
1 c	1 e		8
1 d	1 f		8
1 e	1 g		8
1 f	1 h		8
1 g	1 i		9
1 h	1 j		9
3 b		Removed from manuscript	
3 f		Removed from manuscript	

REVIEWERS' COMMENTS:

Reviewer #2 (Remarks to the Author):

The authors have responded to the reviewers critiques in a methodical way and provided additional data. The substantial matter remains that the observed changes in T cell subpopulations are very small and the observations were made in a relatively small number of patients. The authors lay out the evidence, analyse it statistically appropriately and they interpret it in a reasonable way. While I do not think the results are compelling in regard to the magnitude of biological effects, I accept there may be an element of subjectivity in making this judgment, as in reality nobody knows what could be a 'threshold' for any changes to be efficacious.

Reviewer #3 (Remarks to the Author):

In the manuscript "Dimethyl fumarate targets IL-17-producing CD8+ T cells to limit autoimmunity" authors describe a novel mechanism by which DMF suppresses MS/EAE: by inhibiting the development of Tc17 cells. The manuscript is well-written, and all questions raised by reviewers were largely answered. However, there is still an unanswered question whether the reduction in Tc17 cells in DMF-treated MS patients have a biological effect as numbers of Tc17 cells in responders and non-responders after DMF treatment are similar and authors' response does not provide a strong basis for a different perspective. There is also a problem with their transcriptome data: RNASeq results in Fig. 2 derive from total CD8+CD45RA- cells of MS patients where Tc17 account for 1-3% of these cells; thus, analyses/contrast between Tc17 and Teff signaling pathways does not seem accurate in samples with a ~97% contamination of IL-17-negative cells.

Point-by-point reply to Reviewer Comments

Reviewer #2 (Remarks to the Author)

The authors have responded to the reviewers critiques in a methodical way and provided additional data. The authors lay out the evidence, analyse it statistically appropriately and they interpret it in a reasonable way.

We thank the reviewer for appreciation of our work.

The substantial matter remains that the observed changes in T cell subpopulations are very small

We thank the reviewer for this remark. To substantiate our flow cytometry methodology and thereby our data, we include control samples, which we used for validation of patient sample analysis, as Fig. 1 (point-by-point reply) and into the manuscript, as Supplementary Fig. 8. For this, blood from a healthy donor was drawn at a specific time point (January 2018), then processed for PBMC isolation and frozen in several aliquots which served as control samples. The control samples were thawed, rested, restimulated, fixed, stained, acquired and analyzed in 2018 (at the start of responder sample analysis 2018, time point I) and in 2019 at different days (time points II and III, for non-responder analysis) using the same standard protocols, performed by the same person. The analysis reveals a very similar outcome for 2018 and 2019, which gives mean for Tc17: 0.283 ± 0.0153 (\pm sd) (point-by-point reply, Fig. 1, Supplementary Fig. 8, time points I-III). Furthermore, we applied following standardized technical guidelines to assure high reproducibility of patient sample flow cytometry:

- 1. Regular technical maintenance of Aria III to assure high acquisition standard over a long time period.*
- 2. Daily flow cytometry performance checks with cytometer set-up and tracking beads (CS&T) to verify and calibrate instrument settings and performance.*
- 3. Measurement of samples by the same operator and the same machine (Aria III, BD) belonging to the core facility flow cytometry of the University of Marburg.*
- 4. Sample thawing, resting and restimulation was performed using standardized protocols, by the same person using the same reagents. For staining, the same panel with the same antibodies from the same company (Biolegend, see Methods section) was used.*

I would like to stress, that we analyzed blood samples “before” and “after” DMF therapy always on the same day, both for responders and non-responders. This ensured methodological stringency for the detection of DMF therapy effect on Tc17 cells, which avoided possible day-to-day variations or batch-effect in a relative detection of the DMF effect on Tc17 cells within the respective longitudinally analyzed patient (main manuscript Fig. 1a). For responder samples (main manuscript Fig. 1b), there is a relatively high variation in the frequency of Tc17 cells, however this is not caused by measurement on different days, because in some samples there was high, while in other low Tc17 frequency on one day. Therefore, I believe that our controlled and standardized work-flow assured high quality sample processing over time, allowing a detection of biologically significant phenomena in MS even in a small subpopulation. Finally, I would like to mention that the mean frequency of Tc17 cells before therapy was $1.23\% \pm 0.99$, while after $0.51\% \pm 0.42$, meaning that the effect of DMF on Tc17 frequency before versus after the therapy was rather strong, namely factor 2.4. The suppression of Tc17 cells by DMF in responders is further supported by following data:

- Two mouse models, active and adoptive transfer EAE (Fig. 6, Supplementary Fig. 6)
- Mechanistic data on mouse and on human Tc17 cells (Fig. 1, 2c, 3-5, Supplementary Fig. 1-5)
- RNA-Seq from MS patients, who responded to DMF, revealing suppression of Tc17 RNA-profile after approx. one year of DMF therapy (Fig. 2g-j)
- By comparison with a published data set on human Tc17 cells, and enrichment of Tc17 profile in CD8⁺ T cells before versus one year of DMF therapy (Fig. 2k,l)

Fig. 1. Flow cytometry batch effect validation. Blood from a healthy donor was drawn, processed for PBMC isolation and then frozen in several aliquots which served as control samples. The control samples were thawed, rested, restimulated, fixed, stained, acquired and analyzed on three different time points: at the start of responder sample analysis (time point I) and one year later at two different days (time points II and III, for non-responder analysis) using the same standard protocols by the same person. In the top panel, gating for CD8⁺ T cells is depicted, in the middle, gating of memory CD45RA⁻ CD8⁺ T cells and at the bottom, IL-17-positive cells. Arrows indicate the sequential gating strategy. Numbers indicate percentages of gated cells.

and the observations were made in a relatively small number of patients. While I do not think the results are compelling in regard to the magnitude of biological effects, I accept there may be an element of subjectivity in making this judgment, as in reality nobody knows what could be a 'threshold' for any changes to be efficacious.

We thank the reviewer for this remark. To better substantiate patient analysis, we include design of our study: Patients visiting University Medical Centres of Mainz and Marburg between 2014-2017 were offered standard immunotherapies according to national treatment guidelines and followed longitudinally in an observational cohort. Inclusion criteria for the DMF cohort were age > 18 years, the ability to give informed consent and a maximum of one basic immunomodulatory therapy prior to DMF (IFN- β , Glatiramer acetate). Blood was collected immediately prior to treatment initiation and approx. one year later. Patients were followed longitudinally and stratified into two groups according to treatment response, defined as NEDA-3 positive or negative status (NEDA-3: i, no relapses, ii, no sustained disability progression measured with the expanded disability status scale (EDSS) and iii, no

new/enlarging T2-weighted lesions in magnetic resonance imaging (MRI)) under an appropriate treatment period with DMF (re-baselining 4 months after treatment start).

In total, 65 patients were included in the study cohort, comprising 40 responders and 25 non-responders. 3 non-responders were excluded as therapy was escalated due to disease activity after 4 months after treatment initiation (Fig. 2). In agreement with DMF clinical efficacy¹⁻³, more patients were included into the responder group (N=40). To exclude a selection bias and to obtain comparable group sizes, propensity score matching 1:1 to responders and to non-responders based on age, gender, MS duration since diagnosis and EDSS before treatment was performed. If multiple qualified patients were available for matching, random selection was employed. Selected (n = 21) and excluded (n = 19) responders did not differ concerning above mentioned parameters. From the measured 21 DMF-responder samples, 3 samples were excluded, while from 22 non-responder samples, 4 samples were excluded, all because of low cellularity within the primary sample. We include the study design description into the Methods section and the scheme of study design into the manuscript as Supplementary Fig. 7. Based on in vitro experiments we performed power analysis (see Methods section), which revealed that by an available patient pool of n=18 (paired design, assuming normally distributed values), we would expect to obtain a power of about 95%. Therefore, we are convinced that in terms of both statistics and methodology our data reflect biological significance in MS.

Fig. 2. Scheme of study design for collection of DMF responders and non-responders. In total, 65 patients were included in the study cohort, comprising 40 responders and 25 non-responders. 3 non-responders were excluded because of early therapy escalation. Propensity score matching was performed with the variables age, gender, MS duration since diagnosis, and EDSS before treatment to select 21 out of 40 patients yielding in comparable group sizes for responders and non-responders. Selected (n = 21) and excluded (n = 19) responders did not differ concerning above mentioned parameters. From the measured 21 DMF-responder samples, 3 samples were excluded while from 22 non-responder samples, 4 samples we excluded, all because of low cellularity.

Reviewer #3 (Remarks to the Author)

In the manuscript “Dimethyl fumarate targets IL-17-producing CD8+ T cells to limit autoimmunity” authors describe a novel mechanism by which DMF suppresses MS/EAE: by inhibiting the development of Tc17 cells. The manuscript is well-written, and all questions raised by reviewers were largely answered.

We thank the reviewer for appreciation of our work.

However, there is still an unanswered question whether the reduction in Tc17 cells in DMF-treated MS patients have a biological effect as numbers of Tc17 cells in responders and non-responders after DMF treatment are similar and authors’ response does not provide a strong basis for a different perspective.

We thank the reviewer for this interesting comment. There is an extensive heterogeneity in the MS course and pathological features among patients, which indicates distinct effector mechanisms underlying MS⁴. Based on our data, we believe that there is a subset of patients, by which Tc17 cells contribute to the severity of disease. In these patients, Tc17 cells frequencies are relatively increased before, and reduced by DMF therapy, which correlates with clinical stability. Along with this notion, in non-responders MS is mediated by other mechanisms, as they have generally lower Tc17 proportion before therapy, as compared to responders, which is not altered by DMF treatment. After DMF therapy, both groups, responders and non-responders, have low Tc17 cell frequency, as correctly remarked, but there is a different biological outcome, the responder group is clinically stable, while in the non-responder group the disease is progressing, further supporting the idea on different underlying disease mechanisms. Considering that DMF targets immune and non-immune cells by several mechanisms, we are aware that the clinical stability is a result of complex DMF effects on different cell types, including Th17, macrophages, monocytes, dendritic cells, B cells and neurons^{2, 5-8}. However, based on our data, we believe that targeting of Tc17 cells by DMF is an important mechanism contributing to disease stability in a subgroup of patients. A part of this discussion is included into the discussion section of the current version of this manuscript.

There is also a problem with their transcriptome data: RNASeq results in Fig. 2 derive from total CD8+CD45RA- cells of MS patients where Tc17 account for 1-3% of these cells; thus, analyses/contrast between Tc17 and Teff signaling pathways does not seem accurate in samples with a ~97% contamination of IL-17-negative cells.

We thank the reviewer for raising this important question concerning the transcriptome analyses. While we acknowledge it can be difficult to identify signatures for cells not so highly represented, we substantiate our observations with evidence from two other sources: a mouse dataset, included in this manuscript, and a published human dataset.

- 1. To focus on Tc17 versus “CTL-like” profile, we made use of RNA-Seq data sets of in vitro cultured murine Tc17 cells, “with” versus “without” DMF treatment, displayed in Fig. 2a. In a such system, we were able to obtain a higher percentage of Tc17 cells, (as displayed in Fig 1c). From this mouse Tc17 transcriptome data (Fig. 2a), the top differentially expressed (DE) transcripts “with DMF*

treatment” versus “without DMF treatment” were selected using a stringent threshold ($p_{adj} < 0.01$, $|\log_2 Fc| \geq 0.6$) to establish a robust Tc17-specific and DMF-dependent gene expression signature. This enabled GSEA analysis of DE genes in the human data set, which identified relative enrichment in 182 Tc17-specific and DMF-dependent orthologues in treated versus untreated human transcriptome samples. In the legend for Fig. 2h we described the analysis methods in detail. The comparison of the top DE genes in the mouse with the human dataset revealed a similar shift in expression patterns for the majority of genes (depicted in Fig. 2h as heatmaps). Thereby, this analysis confirmed our previous observation (Fig. 2a-e, g) that genes associated with Tc17 cells are highly influenced by DMF therapy, also in memory CD8⁺ T cells from MS patients. Thus, using a robust set of genes which was shown before to be influenced by DMF in a relatively homogeneous murine Tc17 population ($\geq 30\%$), we were able to detect changes in gene expression even in a small (1-3%) population of human Tc17 cells.

2. To further substantiate Tc17 versus “CTL-like” signature, we made use of a recently published data set (GSE96741⁹) of human Tc17 (IL-17⁺CD8⁺) and “CTL-like” (IL-17⁻CD8⁺) cells isolated from healthy human spleen. GSEA revealed that memory CD8⁺ T cells from untreated patients exhibited more similarity to IL-17⁺CD8⁺ T cells (Tc17) than cells from DMF-treated patients, which in turn were more similar to IL-17⁻CD8⁺ T cells (“CTL-like”) (Fig. 2k, l and Supplementary Fig. 2f, g). Herewith we focused on the expression of human Tc17 signature genes and IL-17⁻CD8⁺ (“CTL-like”) signature genes.

Thus, by analysing two additional datasets (mouse and human), and obtaining overall similar biological outcomes, we are convinced that we were able to detect a DMF-dependent shift in Tc17 versus “CTL-like” signature in the transcriptome data from memory CD8⁺ T cells of DMF-treated MS patients.

References

1. Bar-Or A, *et al.* Clinical efficacy of BG-12 (dimethyl fumarate) in patients with relapsing-remitting multiple sclerosis: subgroup analyses of the DEFINE study. *J Neurol* **260**, 2297-2305 (2013).
2. Linker RA, Haghikia A. Dimethyl fumarate in multiple sclerosis: latest developments, evidence and place in therapy. *Ther Adv Chronic Dis* **7**, 198-207 (2016).
3. Gold R, *et al.* Placebo-controlled phase 3 study of oral BG-12 for relapsing multiple sclerosis. *N Engl J Med* **367**, 1098-1107 (2012).
4. Dendrou CA, Fugger L, Friese MA. Immunopathology of multiple sclerosis. *Nat Rev Immunol* **15**, 545-558 (2015).
5. Schulze-Topphoff U, *et al.* Dimethyl fumarate treatment induces adaptive and innate immune modulation independent of Nrf2. *Proc Natl Acad Sci U S A* **113**, 4777-4782 (2016).

6. Kornberg MD, *et al.* Dimethyl fumarate targets GAPDH and aerobic glycolysis to modulate immunity. *Science* **360**, 449-453 (2018).
7. Ghoreschi K, *et al.* Fumarates improve psoriasis and multiple sclerosis by inducing type II dendritic cells. *J Exp Med* **208**, 2291-2303 (2011).
8. Diebold M, *et al.* Dimethyl fumarate influences innate and adaptive immunity in multiple sclerosis. *J Autoimmun* **86**, 39-50 (2018).
9. Mielke LA, *et al.* TCF-1 limits the formation of Tc17 cells via repression of the MAF-ROR γ axis. *J Exp Med*, (2019).